# Structures and mechanism of the plant PIN-FORMED auxin transporter

Kien Lam Ung[1,4], Mikael Winkler[1,4], Lukas Schulz[2], Martina Kolb[2], Dorina P. Janacek[2], Emil Dedic[1], David L. Stokes[3], Ulrich Z. Hammes[2✉] & Bjørn Panyella Pedersen[1✉]

Auxins are hormones that have central roles and control nearly all aspects of growth and development in plants[1–3]. The proteins in the PIN-FORMED (PIN) family (also known as the auxin efflux carrier family) are key participants in this process and control auxin export from the cytosol to the extracellular space[4–9]. Owing to a lack of structural and biochemical data, the molecular mechanism of PIN-mediated auxin transport is not understood. Here we present biophysical analysis together with three structures of *Arabidopsis thaliana* PIN8: two outward-facing conformations with and without auxin, and one inward-facing conformation bound to the herbicide naphthylphthalamic acid. The structure forms a homodimer, with each monomer divided into a transport and scaffold domain with a clearly defined auxin binding site. Next to the binding site, a proline–proline crossover is a pivot point for structural changes associated with transport, which we show to be independent of proton and ion gradients and probably driven by the negative charge of the auxin. The structures and biochemical data reveal an elevator-type transport mechanism reminiscent of bile acid/sodium symporters, bicarbonate/sodium symporters and sodium/proton antiporters. Our results provide a comprehensive molecular model for auxin recognition and transport by PINs, link and expand on a well-known conceptual framework for transport, and explain a central mechanism of polar auxin transport, a core feature of plant physiology, growth and development.

Auxins are a group of hormones that regulate nearly all growth and developmental processes in plants. Indole-3-acetic acid (IAA; $pK_a = 4.7$) is the most prominent auxin, and is synonymously referred to as 'auxin'. IAA provides a growth signal that orchestrates most complex environmental responses in plants, including phototropism and geotropism[1].

Many of the effects on plant growth depend on the distribution of auxin in the plant body, which is controlled by the process of polar auxin transport[2,3]. This process relies on export of auxin out of cells by PIN transporters[4–9]. The physiological importance of PINs is underlined by often severe *pin* mutant phenotypes, which can be mimicked by auxin efflux inhibitors such as the commercially available herbicide naphthylphthalamic acid[10] (NPA (also known as naptalam); $pK_a = 4.6$).

The PIN protein family is exclusive to the plant kingdom and is classified as part of the large bile/arsenite/riboflavin transporter (BART) superfamily, which also includes transporters of bile acid, arsenite and riboflavin with members distributed across all kingdoms of life[11,12]. PIN proteins are predicted to have ten transmembrane helices comprising two five-transmembrane helix repeats separated by a cytosolic loop. Canonical PINs (PIN1–4 and PIN7 in *A. thaliana*) are characterized by a long (323–355 residue) loop and are mostly located in the plasma membrane, whereas non-canonical PINs (PIN5 and PIN8 and the intermediate PIN6 in *A. thaliana*) possess a much shorter loop and can be found in organellar membranes such as endoplasmic reticulum membranes[13–15]

(Extended Data Figs. 1 and 2a). The long loops of canonical PINs have phosphorylation sites that regulate activity; the loops have been shown to be auto-inhibitory, requiring kinase activity to initiate transport[16].

Here we present structural and biophysical characterization of a PIN protein. In particular, cryo-electron microscopy (cryo-EM) has been used to solve structures in an outward-facing state with and without bound IAA as well as in an inward-facing state with bound NPA at resolutions between 2.9 and 3.4 Å. Combined with transport data from mutant protein, these structures suggest a molecular mechanism and model for auxin transport that is broadly applicable to the ubiquitous PIN family.

We chose to study PIN8 from *A. thaliana* after screening various PIN homologues for expression and purification. PIN8 is a non-canonical PIN of 40 kDa in size, with a short cytosolic loop of 43 residues that lacks the phosphorylation motifs seen in the long auto-inhibitory loops of canonical PINs. When expressed in oocytes, PIN8 exhibited robust IAA transport activity similar to that of kinase-activated PIN1. This activity is independent of activating kinases and sensitive to the inhibitor NPA, demonstrating that PIN8 is a constitutively active auxin transporter (Fig. 1a and Extended Data Fig. 2b).

To characterize electrogenic transport of IAA by PIN8, we overexpressed the protein in *Saccharomyces cerevisiae* and, following purification, reconstituted it into proteoliposomes. We measured transport using capacitive coupling using solid supported membrane (SSM) electrophysiology,

[1]Department of Molecular Biology and Genetics, Aarhus University, Aarhus, Denmark. [2]Plant Systems Biology, School of Life Sciences Weihenstephan, Technical University of Munich, Freising, Germany. [3]Skirball Institute of Biomolecular Medicine, Department of Cell Biology, New York University School of Medicine, New York, NY, USA. [4]These authors contributed equally: Kien Lam Ung, Mikael Winkler. ✉e-mail: ulrich.hammes@tum.de; bpp@mbg.au.dk

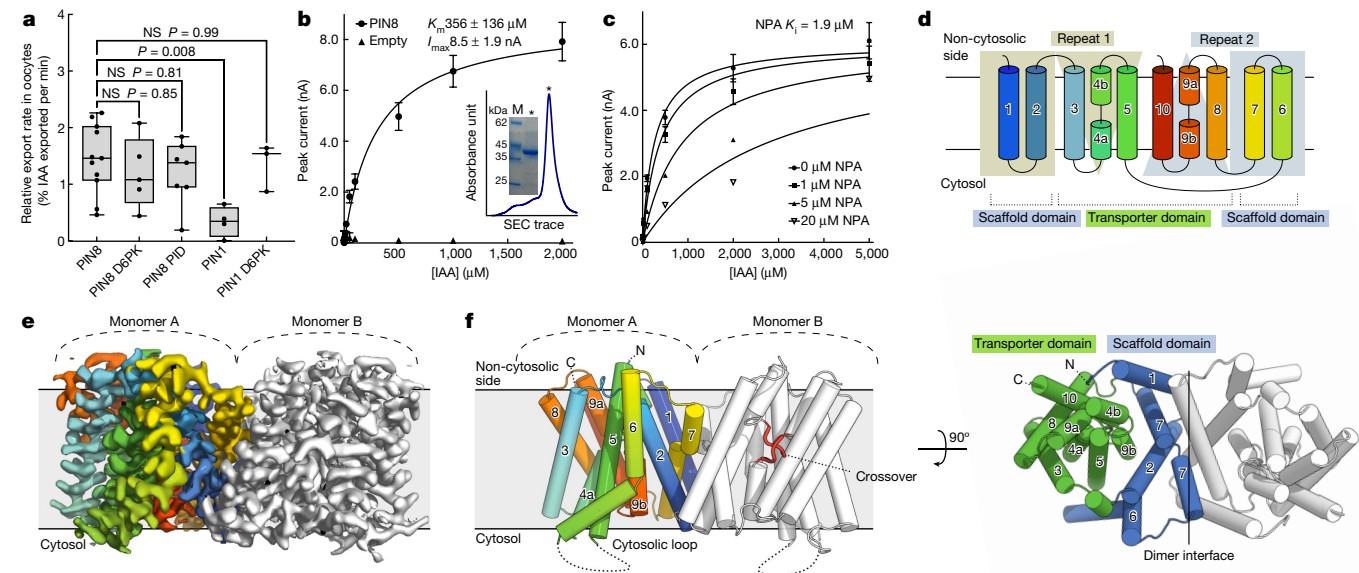

**Fig. 1 | Activity and overall structure of PIN8. a**, Relative IAA transport rates for PIN8 and PIN1 incubated with PIN-activating kinases D6PK and PID show that PIN8 is constitutively active in oocytes (internal oocyte IAA concentration = 1 μM). The centre line is the median, the box extends from the 25th to 75th percentile and whiskers extend to minimum and maximum values. Points represent biologically independent experiments (PIN8: $n = 11$, PIN8 D6PK: $n = 5$, PIN8 PID: $n = 7$, PIN1: $n = 4$, PIN1 D6PK: $n = 3$). For differences between PIN8 and other groups, a one-way ANOVA followed by Dunnett's multiple comparisons test was performed. PIN8 versus PIN8 D6PK: $P = 0.8508$, PIN8 versus PIN8 PID: $P = 0.8090$, PIN8 versus PIN1: $P = 0.0078$, PIN8 versus PIN1 D6PK: $P = 0.9968$. **b**, Peak current response by SSM electrophysiology on PIN8 proteoliposomes. Transport is described by Michaelis–Menten kinetics ($r^2 = 0.98$, $K_m = 356 \pm 136$ μM, maximum current ($I_{max}$) = $8.5 \pm 1.9$ nA; data are mean ± s.e.m.; PIN8: $n = 4$ different proteoliposome preparations, empty: $n = 3$ liposome preparations). Inset, stained SDS–PAGE analysis and size-exclusion chromatography (SEC) trace for the PIN8 purification. **c**, Transport current in the presence of NPA shows inhibition ($K_i = 1.9$ μM (95% confidence interval: 0.9–3.8 μM; $n = 3$ for 0 and 1 μM NPA and $n = 2$ for 5 and 20 μM NPA); data are mean or mean ± s.e.m. ($n > 2$)). **d**, Topology of the PIN8 monomer shows an inverted repeat of five transmembrane helices and the relation between transporter and scaffold domains. **e**, Cryo-EM map of the PIN8 dimer with one monomer coloured according to panel **d**. **f**, Side view of PIN8 with M1–M10 labelled. The central crossover highlighted in red in monomer B. Right, top view from the non-cytosolic side displays the dimer interface and the two domains found in each monomer: the transporter domain (green) and the scaffold domain (blue).

and show that PIN8 has a relatively low apparent affinity for IAA, with a Michaelis constant ($K_m$; Methods, 'SSM physiology assays') of $356 \pm 136$ μM ($n = 4$) (Fig. 1b and Extended Data Fig. 2c). We measure the dissociation constant ($K_d$) of IAA binding to be 39.9 μM (Extended Data Fig. 2d). We observe a modest pH dependence with an optimum at 6.0–7.4 (Extended Data Fig. 2e). As in oocyte assays, transport can be inhibited by NPA, which inhibit with an inhibition constant ($K_i$) of 1.9 μM, suggesting an affinity one order of magnitude higher than that of IAA (Fig. 1c). We screened a number of additional PIN substrates (Extended Data Fig. 2f) and find that IAA analogues—for example, naphthaleneacetic acid (NAA) or the herbicide 2,4-dichlorophenoxyacetic acid (2,4-D), elicit a current response in PIN8, whereas uncharged auxins as well as some endogenous auxins does not. Comparison of these substrates suggests that shape complementary has a large role in recognition: for example, the larger size of indole-3-butyric acid (IBA) and the reduced ring system of 2-phenylacetic acid (PAA) both result in reduced currents.

We solved three distinct structures of PIN8 using single-particle cryo-EM after reconstitution of the purified protein into peptidisc: an apo form at 2.9 Å resolution, PIN8 with IAA bound at 3.2 Å, and PIN8 with NPA bound at 3.4 Å resolution (Extended Data Figs. 3–5 and Extended Data Table 1). In addition, a structure of the apo form that is indistinguishable from the apo peptidisc structure was produced from a detergent-solubilized preparation at 3.3 Å (Extended Data Table 1). The highest-resolution map of the apo form was used for initial model building, but all maps display excellent density for the entire protein except for 39 residues of the disordered cytosolic loop, which were not modelled (Fig. 1d,e). We could model multiple water molecules and lipids as well as IAA and NPA in the relevant structures.

The apo form of PIN8 displays a symmetric dimer of PIN8 (Fig. 1f) characterized by a twofold rotation axis perpendicular to the membrane

plane with a distinct concavity extending into the membrane along this axis from the non-cytosolic side. Within each monomer there are ten transmembrane helices (M1–M10), comprising an inverted repeat of five transmembrane helices[17] (Fig. 1d). In each repeat, the fourth helix is disrupted around a conserved proline residue in the middle of the membrane plane: Pro116 in M4 and Pro325 in M9. These disrupted helices make an X-shaped crossover that marks the auxin binding pocket (Fig. 1f).

The PIN8 monomer is divided into two domains that we name the scaffold domain and the transporter domain (Fig. 1d,f and Extended Data Fig. 6a). The scaffold domain comprises helices M1, M2, M6 and M7 and creates a large interface (1,512 Å²) to the other monomer in the dimeric complex. This interface is mediated mainly by M2 and M7, and is further stabilized by a lipid in a groove between M1 and the kinked M6 (Extended Data Fig. 6b). We also observe another lipid with an aliphatic tail sticking into a pocket of the transporter domain. We tested a dependence on lipids for activity and found that PIN8 functions similarly in mixed lipid and pure phosphatidylcholine liposomes (Extended Data Fig. 6c). The transporter domain consists of helices M3–M5 and M8–M10 and harbours the central X-shaped crossover (Fig. 1f). The overall fold of the monomer is similar to that of the bile acid/sodium symporters, but the membrane topology is inverted[18] (Extended Data Fig. 7). Next to the crossover, there is a well-defined water-filled binding pocket nestled between the scaffold domain and the transporter domain that is open to the non-cytosolic side of the protein via the concavity (Fig. 1f). By contrast, access to the cytosol is blocked, clearly defining the conformation of the apo-PIN8 dimer as an empty outward-open state.

The substrate-bound form of PIN8, IAA–PIN8, is almost identical to apo-PIN8 (root mean squared deviation of Cα atoms (r.m.s.d._Cα) = 0.6 Å)

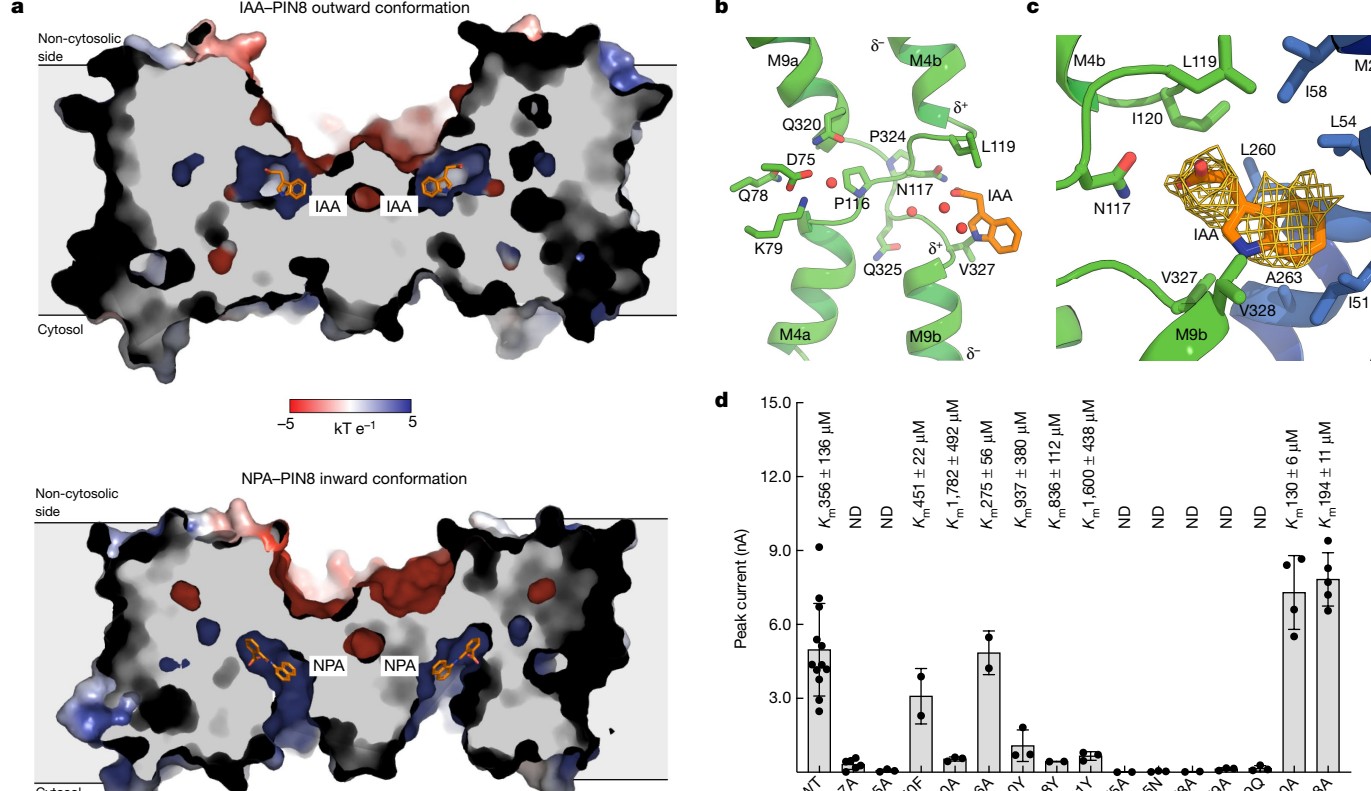

**Fig. 2 | Structures with IAA and NPA bound. a**, Cutaway view of electrostatic surface representation of IAA–PIN8 and NPA–PIN8 show the change in conformation. Whereas the concavity at the non-cytosolic side has negative potential, the binding pocket itself has a positive potential in both cases. **b**, View of the crossover and the position of IAA and the support site with central residues highlighted. **c**, Close-up view of IAA map density and the residues from the scaffold domain interacting with the indole ring. **d**, Peak current response evoked by 500 μM IAA determined by SSM electrophysiology

for PIN8 mutants. $K_m$ values (mean ± s.d.) derived from the full measurement (Extended Data Fig. 8a) are shown above the bars; ND indicates that a Michaelis–Menten curve could not be fit. The bars show mean ± s.e.m. ($n$ > 2); points represent biologically independent measurements (wild type (WT): $n$ = 4 different liposome preparations; mutants: $n$ = 6 (N117A), $n$ = 5 (T288A), $n$ = 4 (Q320A), $n$ = 3 (Q145A, D75N, K79A, K79Q, Q78A, I51Y, I120Y and Y150A), $n$ = 2 (Y150F, S146A, D75A and V328Y)).

(Fig. 2a). There is a clear density for IAA in the binding pocket, with three surrounding water molecules (Fig. 2b,c). Thus, the IAA–PIN8 structure represents a substrate-bound outward-open state, the expected release state for auxin.

IAA is bound with its carboxylate group oriented towards the crossover; although only two residues are within hydrogen-bonding distance (Asn117 and Gln145), IAA is stabilized by the positive dipole from M4b and M9b helices. The backbone carbonyl of Pro116 creates a polar pocket that is also lined by Tyr150 and Ser146. Here we observe three well-defined water molecules that may reflect partial disassociation of IAA from the binding pocket in the release state. Mutating either Asn117 and Gln145 to alanine abolishes transport, supporting their importance (Fig. 2d). Tyr150 mutants display mixed results: Y150F retains activity, affinity and sensitivity to NPA, whereas removal of the bulky side chain in Y150A results in very low activity and affinity (Extended Data Fig. 8a,b). By contrast, mutation of Ser146 had no effect on activity (Fig. 2d and Extended Data Fig. 8a).

In the transporter domain, the IAA carbon backbone contacts Leu119(M4b) and Ile120(M4b) towards the non-cytosolic side, whereas the indole ring contacts Val327(M9b) and Val328(M9b) towards the cytosolic side. These four hydrophobic residues are symmetrically located on the crossover immediately after the two key prolines as part of a duplicated and conserved crossover sequence motif (P(N/Q) XΦΦ; where Φ is a hydrophobic residue) (Fig. 2b–d and Extended Data Figs. 1 and 8b). The hydrophobic residues of the crossover motif provide affinity for the auxin substrate. This is supported by the bulky I120Y and

V328Y mutants, which both reduce apparent affinity by interfering with substrate binding but still retain NPA sensitivity (Fig. 2d and Extended Data Fig. 8a,b). Together, the interactions between the transporter domain and IAA emphasize that PIN8 selects for IAA on the basis of shape complementarity, as also suggested by the SSM electrophysiology results. In the scaffold domain, the indole ring has additional non-specific hydrophobic interactions with Ile51 (M2), Leu54 (M2) and the pseudo-symmetrically related Leu260 (M7) and Ala263 (M7). Bulky mutations in these hydrophobic residues (such as I51Y) lead to a considerably reduced transport current (Fig. 2d and Extended Data Fig. 8a,b). All the residues defining the binding pocket show high sequence conservation across different plant species and are fully conserved in all *A. thaliana* PIN proteins except PIN5 (Extended Data Figs. 1 and 8c).

The NPA-bound form of PIN8 adopts an inward-open conformation (Fig. 2a). The scaffold domains and dimeric interface is unchanged relative to the outward-open conformation (r.m.s.d.$_{Cα}$ = 0.9 Å), but the two transporter domains are rotated by approximately 20° to expose the auxin binding site and Asn117 to the cytosolic side. This rotation results in a translation of the binding site by approximately 5 Å (Fig. 3a and Supplementary Video 1). NPA has more extensive interaction with the protein compared with IAA in the outward-open state (Fig. 3b,c and Extended Data Fig. 8c,d). Similar to IAA, the carboxylate group of NPA points towards the crossover, but has several stronger interactions that are not observed in the outward-open state. In addition to interactions seen for IAA, NPA interacts with main chain nitrogen atoms of Val327

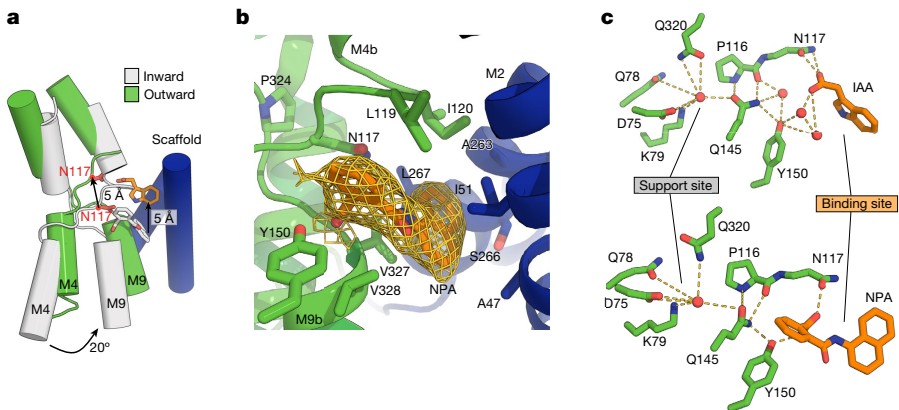

**Fig. 3 | Conformational change and the support site. a**, Inward and outward structures superposed on the scaffold domain (blue) reveal an elevator-type movement, with the substrate binding site moving 5 Å. **b**, Close-up view of the NPA map density and the residues interacting with it. **c**, The hydrogen-bonding network linking the binding site to the support site through Gln145 in the outward state (top) and inward state (bottom).

and Val328, as well as with Gln145 and Tyr150 in a network that does not involve water. The benzene ring and naphthyl ring of NPA still interact with the two crossover motifs of the transporter domain, similar to IAA. Several new interactions are also observed with the scaffold domain, many of which are mediated by the naphthyl ring of NPA and are probably unique to the larger, more complex NPA molecule (Fig. 3b,c and Extended Data Fig. 8a–d). Inhibition by NPA can thus be explained by two components: (1) stronger binding due to engagement of additional residues from the scaffold domain, and (2) the larger size of NPA that prevents transition to the outward state.

Adjacent to the primary auxin binding site, an accessory 'support site' is apparent on the other side of the crossover between M3, M5 and M9. This support site is linked to the primary auxin binding site via an extensive hydrogen bond network bridged by the central Gln145 and the backbone carbonyl of Pro116. The higher-resolution apo-PIN8 map reveals two peaks in the site, which are modelled as water (Extended Data Fig. 8e). In the lower-resolution IAA–PIN8 and NPA–PIN8 map, the same site contains one single weak peak that is also modelled as water (Fig. 3c and Extended Data Fig. 8c–e). The presence of Na$^+$ at analogous sites in bile acid/sodium symporters led us to probe ion dependence by comparing PIN8 transport in sodium- and potassium-exclusive buffers. In both cases, PIN8 retains full activity, suggesting that specific counter-transport of ions does not take place (Extended Data Fig. 9a). In all structures, the water molecules in the support site engage in a hydrogen bond network with Asp75 (M3), Gln78 (M3), Lys79 (M3), Gln320 (M9a) and Gln145 (M5) (Fig. 3c). Mutational analysis indicates that all of these residues except Gln320 are absolutely essential for activity (Fig. 2d and Extended Data Fig. 8a). Notably, Asp75 and Lys79 are fully conserved and constitute a proton donor–acceptor pair with potential for proton transport; indeed, this idea is supported by isosteric mutations that remove the charge from either residue (D75N or K79Q) and abolish transport (Fig. 2d and Extended Data Fig. 8a,c,d). The distance from Asp75 to Lys79 is below 3 Å in all structures, consistent with an unprotonated state for Asp75. However, activity of PIN8 in proteoliposomes is not sensitive to proton-motive force decouplers and has minimal pH dependence, suggesting that a proton-motive force is not obligatory for transport (Extended Data Figs. 2e and 9b). Furthermore, export rates in oocytes are also indifferent to external pH (Extended Data Fig. 9c).

## Discussion

Plant growth and morphology are largely governed by polar auxin transport as mediated by canonical PINs. Comparison of all the PINs from *A. thaliana* with PIN8 studied here indicates that—with an exception of the unusual non-canonical PIN5—the auxin and support sites are perfectly conserved. This conservation, which also extends to other plant species, indicates that our observations can be generalized[19] (Extended Data Figs. 1, 2a and 8c,d). The low apparent affinity for IAA measured in proteoliposome assays is 5–500-fold lower than the physiological concentrations of auxin in plant tissues[20] (0.1–10 µM). Although we cannot rule out experimental artifacts, this implies that distinct functions of *A. thaliana* PINs arise from differing localization, abundance and auto-inhibition properties rather than direct modulation of substrate affinity[3]. Some studies have suggested that ABCB transporters interact with PINs to generate selectivity in IAA transport[21–23]. Our work suggests that this interaction is not needed for activity in vitro, and is most probably not required in planta.

The PIN family is part of the BART superfamily, which includes the structurally characterized ASBT bile acid/sodium symporters from the BASS family[18]. Although PIN8 and ASBT adopt the same fold, ASBT assumes an inverted orientation and does not appear to dimerize (Extended Data Fig. 7). In addition, at least three other families of proteins adopt this same fold (DALI Z-score > 10), namely two Na$^+$/H$^+$ antiporter families (CPA1 and CPA2) and the HCO3$^-$/Na$^+$ symporter family[24–29]. Similar to the bile acid/sodium symporters, these other protein families all share negligible sequence homology with PINs. The HCO3$^-$/Na$^+$ symporters adopt the same membrane orientation as PIN8, whereas the Na$^+$/H$^+$ antiporters share the inverted orientation with the bile acid/sodium symporters, perhaps explaining why the structural link between PINs and these divergent protein families has not been noted previously (Extended Data Fig. 7).

These other protein families are all secondary active transporters that use sodium or protons to drive transport, and all are proposed to function using an elevator mechanism in which the scaffold domain is fixed and the transporter domain pivots about the conserved proline crossover motif. Notably, the site occupied by the driving sodium and protons in these families is located at the same position as the support site in PINs (Extended Data Fig. 7), and it is clear from this work that PIN8 uses the same general proline crossover-based elevator mechanism (Fig. 3a and Supplementary Video 1).

Our data show that the negative charge of the IAA is sufficient for transport (Extended Data Figs. 2e and 9). However, the basic architecture of a support site is present that would allow for ion binding, as well as a conserved and functionally essential Asp75–Lys79 pair that could mediate proton translocation[30]. Most mutations of the support site completely abrogate activity, underlining the essential nature of this region, but neither oocyte nor SSM electrophysiology assays suggest dependence on counter-transport of either sodium or protons to drive auxin export. Our data thus support a uniport mechanism for PINs, although we cannot definitely rule out proton antiport in vivo.

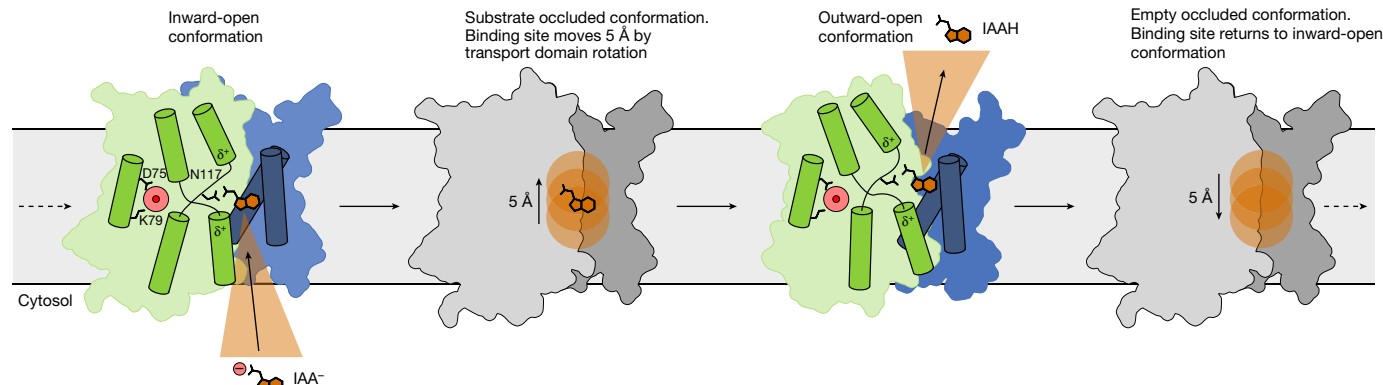

**Fig. 4 | Proposed mechanism of auxin export by PIN proteins.** In the inward-open conformation (left), IAA enters the binding site with a deprotonated carboxylate. The positive dipole of the M4b and M9b helices helps diffuse the charge. During rotation of the transporter domain, the binding site moves 5 Å towards the non-cytosolic side (second from left). At the non-cytosolic side, IAA is released, probably assisted by a protonation event (second from right). The support site could also become protonated before reverting to the inward-facing conformation (right), but our data indicate that this is not obligatory for function.

On the basis of the data available, we propose the following model for auxin transport by PINs (Fig. 4): The inward-facing conformation allows an ionized auxin molecule to enter the binding site between transport and scaffold domains. The negatively charged carboxylate group is stabilized by the positive dipole of M4b and M9b, while being held in place by Asn117 and interacting with the support site through Gln145. The carbon backbone and indole ring are recognized by the four hydrophobic residues from the two crossover motifs of the scaffold domain. During transition to the outward-facing conformation, the proline crossover rotates 20° and the auxin binding site in the scaffold domain is translated away from the cytosol by 5 Å. Release of IAA in the outward-facing state is facilitated by a pH shift that protonates and neutralizes the carboxylate. After substrate release, the protein reverts back to the inward-open state.

It has been suggested that the oligomeric state of PINs might have a role in regulation, but the large dimer-interaction surface in PIN8 argues against a dynamic equilibrium[31,32]. Nevertheless, it is conceivable that the monomers operate independently and also that PINs could form hetero-oligomers[33].

We have not directly addressed auto-inhibition by the cytosolic loop in canonical PINs, but the connection to other known protein families provides some hints: For $HCO_3^-/Na^+$ symporters, it has been shown that a loop from a cytosolic regulatory partner locks the protein in an inward conformation by interacting with the binding site[24,29]. By analogy, it seems plausible that the auto-inhibitory loop in canonical PINs operates by a similar mechanism.

In conclusion, we have presented in vitro biochemical characterization of a PIN as well as structures representing two key conformational states in the presence and absence of auxin and the herbicide NPA. The structure with NPA demonstrates competitive inhibition in PIN proteins, and could provide the basis for structure-based development of novel herbicides. We describe the molecular mechanism of auxin transport by PINs that can function independently of monovalent ions or protons, thus expanding our understanding of the crossover elevator mechanism used by proteins from diverse protein superfamilies from all kingdoms of life. This work provides a comprehensive foundation for future studies aiming to elucidate PIN function in polar auxin transport, which is essential for plant growth and development.

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

# Methods

## Protein purification

*A. thaliana* protein sequences used in this study are publicly available at Uniprot (https://www.uniprot.org/) with the following accession codes. PIN1: Q9C6B8, PIN2: Q9LU77, PIN3: Q9S7Z8, PIN4: Q8RWZ6, PIN5: Q9FFD0, PIN6: Q9SQH6, PIN7: Q940Y5 and PIN8: Q9LFP6.

PIN genes were cloned into an *S. cerevisiae* overexpression plasmid based on p423_GAL1 and tested for expression and purification properties. The *A. thaliana PIN8* gene (Uniprot: Q9LFP6) was selected and put in frame with a tobacco etch virus (TEV) protease cleavage site and a deca-histidine affinity tag. This construct was used as the template for site-directed mutagenesis using the Quickchange commercial protocol (Agilent) for all point mutants.

Transformed *S. cerevisiae* strain *DSY-5* were grown in 5 l shaking flasks or culture vessels, grown to high cell density and collected after 22 h induction with galactose[34]. Collected cells were washed three times in water and re-suspended in buffer A (0.1 M Tris pH 7.5, 0.6 M NaCl, 1 mM ethylenediamine tetraacetic acid (EDTA), 1.2 mM phenylmethylsulphonyl fluoride). Cells were lysed by bead beating and lysate was clarified by centrifugation at 5,000g for 20 min. Membrane fractions were pelleted by ultracentrifugation at 200,000g for 2 h and re-suspended in buffer B (0.05 M Tris pH 7.5, 0.5 M NaCl, 20% glycerol) before being frozen in liquid nitrogen.

For protein purification, 3–4 g of membrane was thawed and solubilized for 45 min in a total volume of 50 ml of buffer C (0.05 M Tris pH 7.5, 0.5 M NaCl, 10% glycerol) supplemented with 1% *n*-dodecyl-*β*-d-maltoside (DDM) and 0.1% cholesterol hemisuccinate (CHS). Insoluble material was discarded by centrifugation at 17,000g for 30 min following by filtration using a 1.2 μm filter. 20 mM imidazole pH 7.5 was added and the sample loaded on a 1 ml nickel-nitrilotriacetic (Ni-NTA) column. A two-step wash was performed with buffer D (buffer A with 20 mM imidazole, 0.1% DDM, 0.01% CHS) and buffer E (buffer A with 70 mM imidazole, 0.05% DDM, 0.005% CHS).

For SSM electrophysiology assays, the sample was eluted with buffer F (0.05 M Tris pH 7.5, 0.15 M NaCl, 10% glycerol, 0.05% DDM, 0.005% CHS, 500 mM imidazole). The eluate was incubated with TEV protease and dialysed against buffer F supplemented with 0.5 mM EDTA and 0.5 mM tris(2-carboxyethyl)phosphine (TCEP) overnight. The sample was then filtered and re-run on a Ni-NTA column to adsorb the His-tagged proteins consisting of TEV protease, cleaved tag and uncleaved tagged protein. The flow-through fraction, containing tag-free PIN8, was concentrated on a 100 kDa cut-off centricon (Vivaspin) and polished by SEC on a Biorad650 or Superdex200 10/300 column pre-equilibrated with buffer G optimized by a thermostability assay[35] (0.05 M Tris pH 7.5, 0.15 M NaCl, 10% glycerol, 0.05% DDM, 0.005% CHS, 0.5 mM EDTA).

For cryo-EM, peptidisc sample preparation followed general protocols[36,37]. In brief, after the two-step wash, proteins were re-lipidated using buffer I (0.05 M Tris pH 7.5, 0.15 M NaCl, 10% glycerol, 0.03% DDM, 0.003% CHS, 0.06 mg ml$^{-1}$ soybean extract polar lipids (Avanti)). Prior to starting the on-bead peptidisc reconstitution, the column was washed with buffer J (0.05 M Tris pH 7.5, 0.15 M NaCl, 10% glycerol, 0.008% DDM, 0.0008% CHS). Peptidisc reconstitution was initiated by washing the column with detergent-free buffer K (0.05 M Tris pH 7.5, 0.15 M NaCl, 10% glycerol) containing 1 mg ml$^{-1}$ peptidisc (Genscript). An additional washing step with buffer K was performed to eliminate residual free peptidisc prior to elution using buffer K supplemented with 500 mM imidazole. After this the sample was incubated with TEV protease and dialysed against buffer K supplemented with 0.5 mM EDTA and 0.5 mM TCEP.

For the cryo-EM detergent sample, immediately after the re-lipidation step with buffer I, the DDM detergent was exchanged to lauryl maltose neopentyl glycol (LMNG) using buffer L (0.05 M Tris pH 7.5, 0.15 M NaCl, 10% glycerol, 0.006% LMNG, 0.0006% CHS) prior to protein elution using buffer L supplemented with 500 mM imidazole. The sample was then incubated with TEV protease and dialysed against buffer L supplemented with 0.5 mM EDTA and 0.5 mM TCEP. After dialysis, cryo-EM sample purification continued identically to the SURFE²R sample protocol described in 'SSM electrophysiology assays', with the exception that the SEC buffer was replaced with buffer K (peptidisc sample) or buffer L (LMNG sample) without glycerol and supplemented with 0.5 mM EDTA.

## SSM electrophysiology assays

For SSM electrophysiology, a SURFE²R N1 from Nanion Technologies was used. In brief, Soy Phospholipid Mixture (38% phosphatidylcholine, 30% phosphatidyl ethanolamine, 18% phosphatidyl inositol, 7% phosphatidic acid and 7% other soy lipids) and 1-palmitoyl-2-oleoyl-*sn*-glycero-3-phosphocholine (POPC) were purchased from Avanti. Liposomes were prepared in Ringer solution without Ca²⁺ (115 mM NaCl, 2.5 mM KCl, 1 mM NaHCO₃, 10 mM HEPES pH 7.4, 1 mM MgCl₂) and homogenized using a Lipsofast (Avestin Inc) with a 400 nM pore size. Triton X-100 was added to the liposomes to a final concentration of 1% (v/v). Protein was added to liposomes to a calculated liposome:protein ratio (LPR) of 10:1. The detergent was removed using 400 mg ml$^{-1}$ Bio Beads (BioRad) overnight at 4 °C in a rotary shaker. Proteoliposomes were frozen in liquid nitrogen and kept at −80 °C until use.

Sensor coating was performed as described[38]. Proteoliposomes were diluted 1:5 in Ringer solution without Ca²⁺, sonicated five times and then applied to the sensors by centrifugation (30 min, 3,000g, 4 °C). Non-activating buffer was Ringer solution without Ca²⁺ as described unless specified otherwise and activating buffer contained the substrate of interest. To substitute Na⁺, K⁺-Ringer without CaCl₂ (117.5 mM KCl, 10 mM HEPES pH 7.4, 1 mM MgCl₂) and to substitute K⁺, Na⁺-Ringer without CaCl₂ (117.5 mM NaCl, 10 mM HEPES pH 7.4, 1 mM MgCl₂) were used. Uncouplers: carbonyl cyanide *m*-chlorophenyl hydrazone (CCCP) in ethanol was used at 5 μM and 2,4-dinitrophenol (DNP) in ethanol was used at 100 μM. All other chemicals were purchased from Roth or Sigma. Each experiment was performed on at least two individual sensors. On each sensor each measurement consists of three technical replicates where the mean is calculated.

In most instances, we used a single solution exchange experiment. In this case proteoliposomes, immobilized on the supported membrane are kept in non-activating buffer as specified. At the beginning of the experiment non-activating buffer was exchanged for fresh identical non-activating buffer and after 1 s activating buffer (same buffer containing substrate) was added. After a further 1 s, buffer was again exchanged to non-activating buffer. Current response was recorded throughout the entire 3 s. For competition or inhibition, the respective compound was present in non-activating and activating solution.

Currents in response to substrate in the activating solutions are responses to electrogenic events which occur (1) when a charged molecule is crossing the membrane; (2) when a substrate, which does not necessarily have to be charged, binds to the protein and this binding leads to a conformational change by which charges become displaced in the membrane; (3) currents are shielded or neutralized by the substrate; and (iv) any combination of these possibilities. The peak current in response to substrate application was used to describe the properties of the proteins.

To describe the current response to different substrate concentrations a Michaelis–Menten curve was fit. We use $K_m$ throughout the manuscript, but since the peak current is a mixture of binding and transport signal (that is, pre-steady state and steady state currents), this parameter can also be more appropriately described as $EC_{50}$. A $K_m$ derived from a biophysical assay will be specific to that experimental setup, and comparison to other types of assay or a physiological condition should be done cautiously. In the case of competitive studies, we explicitly use $K_d$ or $K_i$, since in these instances the parameters were specifically determined. GraphPad Prism V 9.3 was used for statistical analyses.

### Cryo-EM sample preparation

Peak fractions of freshly purified PIN8 were concentrated to 4–10 mg ml$^{-1}$. C-flat Holey Carbon grids (CF-1.2/1.3, Cu-300 mesh) were glow-discharged for 45 s at 15 mA in a GloQube Plus (Quorum). A drop of 4 µl of sample was applied to the non-carbon side of the grids, and blotted with a Vitrobot Mark IV (ThermoFisher Scientific) operating at 4 °C and 100% humidity and using blot time of 4 s, before plunge-freezing into liquid ethane. The substrate-bound states were obtained by incubating the sample with 15 mM of IAA sodium salt or 2 mM of NPA for 2 h prior to grid freezing.

### Image collection and data processing

A Titan Krios G3i microscope (ThermoFisher Scientific) operating at 300 kV and equipped with a BioQuantum Imaging Filter (energy slit width of 20 eV) with a K3 detector (Gatan) was used to collect the movies. The datasets containing the peptidisc samples, were acquired using automated acquisition EPU v2.11.1.11 at nominal 130,000 magnification corresponding to a physical pixel size 0.647 Å. For all datasets, the movies were saved in super-resolution pixel size and binned 2× in EPU back to the nominal pixel size.

On-the-fly gain normalized exposures were imported into cryoSPARC (v3.2.0)[39] and processed in streaming mode for patch motion correction, patch contrast transfer function (CTF) estimation, particle picking and extraction. After several rounds of particle cleaning, an initial preliminary volume map was used to create templates for template picking. From a full dataset of apo-PIN8 with 7,900 movies, template picking provided a total of 2,082,448 particles. After two rounds of 2D classification, the best representative classes were selected manually. These particles served as an input for ab initio model reconstruction. After three rounds of particle sorting by heterogenous refinement using the ab initio 3D template, the remaining 327,193 particles were used for non-uniform refinement with C2 symmetry imposed and resulted in a global 2.9 Å resolution map. In parallel a C1 symmetry refinement job was performed but showed no differences between the two monomers.

To ensure the method of membrane protein stabilization did not influence oligomeric state and overall structure we solved apo PIN8 both in peptidisc (2.9 Å) and in the detergent LMNG (3.3 Å). The respective maps reveal no variation in conformation and we focus on the peptidisc-derived map given its higher resolution. There was also no evidence of monomers or higher oligomeric states in any of the grids screened.

The processing pipeline for the ligand-bound PIN8 was identical to the one from apo-PIN8. In brief, the entire IAA–PIN8 dataset comprised of 15,771 movies and template picking yielded a total of 2,639,895 particles. After several rounds of 2D classification and heterogenous refinement to obtain a final 200,061 particles, a non-uniform refinement with C2 symmetry imposition resulted in a global 3.2 Å resolution map. A full dataset of NPA–PIN8 comprised of 14,500 movies and template picking yielded a total of 3,345,146 particles. After several rounds of 2D classification and heterogenous refinement to obtain a final 77,608 particles, a non-uniform refinement with C2 symmetry imposition resulted in a global 3.4 Å resolution map. As for the apo form, a parallel C1 refinement was performed with no differences evident between the two monomers. Local resolution estimation was performed using cryoSPARC.

### Model building and refinement

A PIN8 model prediction was calculated using the RoseTTAFold server[40] and docked into the PIN8 map in Chimera[41]. Two molecules of PIN8 could be readily fitted into the map. The flexible cytoplasmic loop of PIN8 (residues 165 to 205) is not visible in the maps and was excluded from model building in Coot[42]. The final models include residues 1–164 and 206–367 (of 367 residues total). The initial PIN8 dimer model was analysed by molecular dynamics-based geometry fitting to the map using MDFF[43] through Namdinator v2.0 (ref. [44]). Models could be further improved by iterative manual model building in Coot combined with real-space refinement using Phenix, initially with an Amber force-field molecular dynamic refinement[45]. The coordination of lipids and the ligand IAA was prepared using ligand builder eLBOW[46]. In all electron microscopy maps, although the lipid belt surrounding the PIN8 dimer is visible, the electron density only allowed for the tentative modelling of two phosphatidylcholine molecules for ligand-bound PIN8 and four molecules for apo-PIN8. Geometry was validated in MolProbity v4.2 including CaBLAM and Ramachandran-Z analysis[47–49] (Rama-Z). Figures were prepared using PyMOL Molecular Graphics System v1.5.0.4 (Schrödinger). Conservation of residues across species was analysed using ConSurf[50]. Sequence alignments were constructed with PROMALS3D[51]. Alignments were visualized using ALINE v1.0.025[52]. Structural similarity to other protein families were identified using DALI[53]. Phylogenetic analysis was made using NGPhylogeny.fr[54]. In brief, MAFFT was used for multiple sequence alignment (MSA), BMGE was used for MSA pruning and FastME was used for unrooted tree generation. Bootstrap values were calculated from 500 trials.

### Oocyte efflux assays

Oocyte efflux experiments were carried out as described[55]. In brief, oocytes were injected with 150 ng transporter cRNA without or with 75 ng kinase cRNA. $^3$H-IAA (25 Ci mmol$^{-1}$) was purchased from ARC or RC Tritec. Oocytes were injected with IAA to reach an internal IAA concentration of 1 µM, corresponding to 100%. Residual radioactivity was determined for each individual oocyte by liquid scintillation counting after the time points indicated and are expressed relative to the initial 100%. Each time point represents the mean and s.e.m. of ten oocytes. To calculate the relative transport rate in per cent per minute, linear regression was performed. Each data point in Fig. 1a and Extended Data Fig. 9c represents the transport rate of one biological replicate using oocytes collected from different *X. laevis* females. GraphPad Prism V 9.3 was used for statistical analyses.

### Reporting summary

Further information on research design is available in the Nature Research Reporting Summary linked to this paper.

## Data availability

Atomic models have been deposited in the Protein Data Bank (PDB) and cryo-EM maps have been deposited in the Electron Microscopy Data Bank (EMDB) under the following accession numbers. Apo outward state in peptidisc: PDB 7QP9 and EMDB EMD-14115, IAA-bound outward state in peptidisc: PDB 7QPA and EMDB EMD-14116, NPA-bound inward state in peptidisc: PDB 7QPC and EMDB EMD-14117, and apo outward state in detergent: EMDB EMD-14118. Source data are provided with this paper.

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

**Acknowledgements** We acknowledge the EMBION Cryo-EM Facility at iNANO, Aarhus University, for time under application ID 0137, where all data was collected with the assistance of A. Bøggild, J. Lykkegaard Karlsen and T. Boesen. We also thank eBIC (proposal BI27980) for data collection on the detergent PIN8 sample. This project has received funding from the European Research Council (ERC) under the European Union's Horizon 2020 research and innovation programme (grant agreement no. 101000936) to B.P.P. U.Z.H. is funded by the Deutsche Forschungsgemeinschaft (HA3468/6-1 and HA3468/6-3) and SFB924. D.L.S. is funded by the National Institutes of Health (R35 GM144109).

**Author contributions** Sample preparation: K.L.U. and M.W. Electron microscopy data collection and analysis: K.L.U., M.W., E.D., D.L.S. and B.P.P. Activity assays: D.P.J., L.S., M.K. and U.Z.H. Manuscript preparation: K.L.U., M.W., D.L.S., U.Z.H. and B.P.P.

**Competing interests** The authors declare no competing interests.

**Additional information**
**Correspondence and requests for materials** should be addressed to Ulrich Z. Hammes or Bjørn Panyella Pedersen.

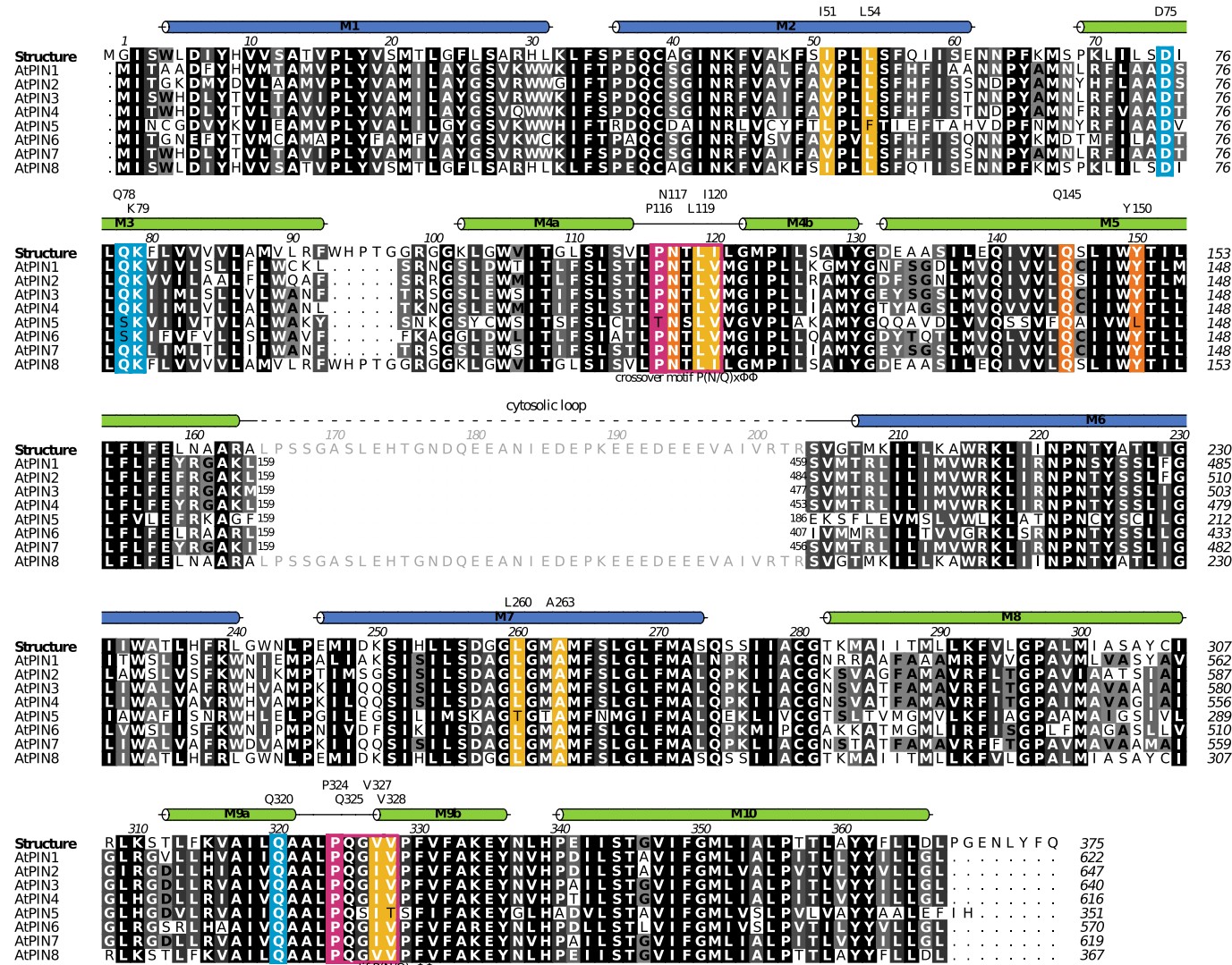

**Extended Data Fig. 1 | Multiple sequence alignment *A. thaliana* PINs.**
Alignment between AtPIN1–8 with the following UniProt accession numbers.
AtPIN1: Q9C6B8, AtPIN2: Q9LU77, AtPIN3: Q9S7Z8, AtPIN4: Q8RWZ6, AtPIN5:
Q9FFD0, AtPIN6: Q9SQH6, AtPIN7: Q940Y5, AtPIN8: Q9LFP6. Conserved
residues are highlighted with gray-scale, where black is perfectly conserved.

Colored tubes represent α-helices found in the scaffold domain (blue) and
transporter domain (green). Key residues are numbered above the α-helix
markings. Residues highlighted participate in IAA carboxylate recognition
(orange) or IAA hydrophobic recognition (yellow), are part of the support site
(blue) or form the central prolines of the crossover motif (pink).

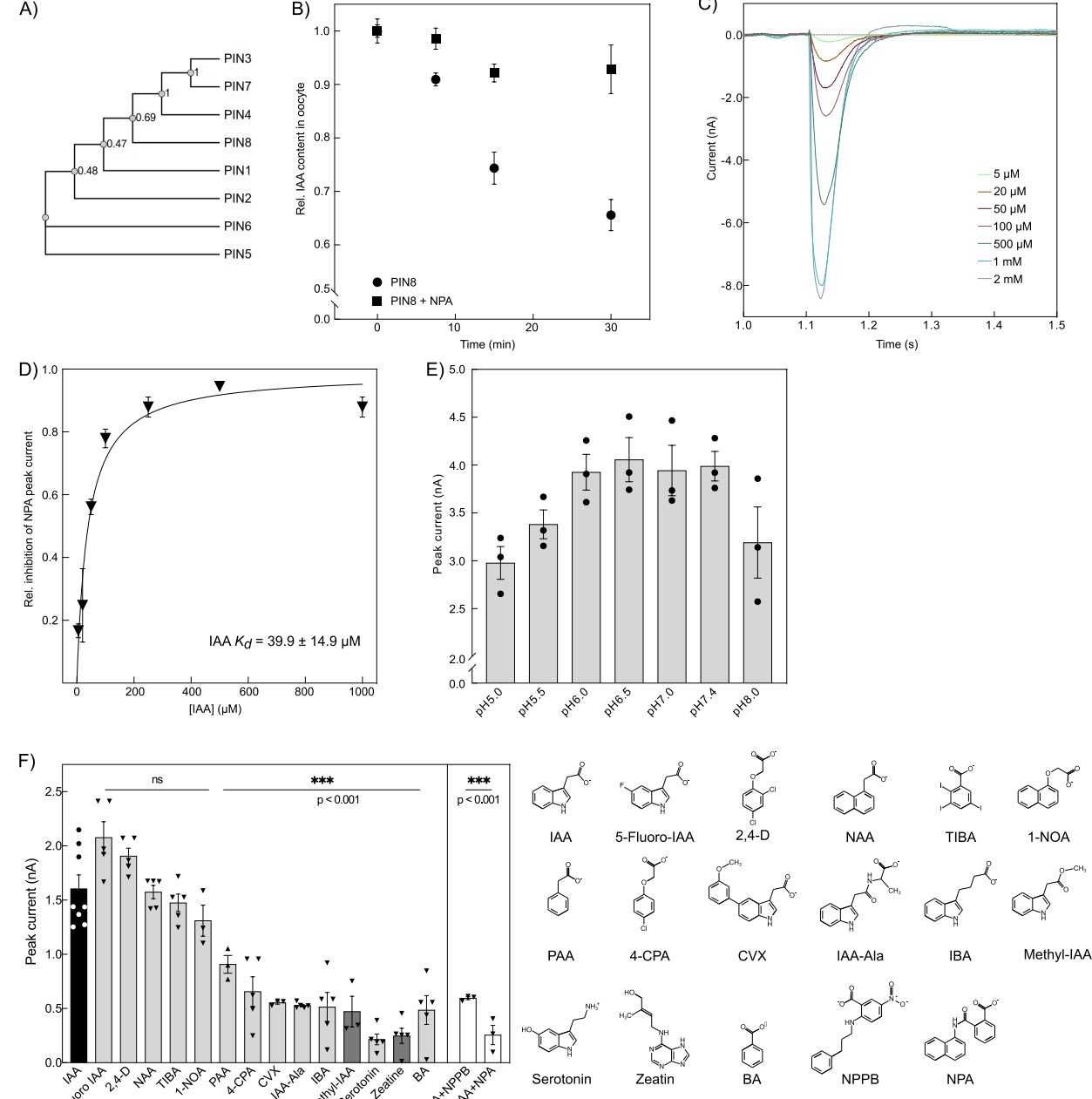

**Extended Data Fig. 2 | Functional data on PIN8. A)** Dendrogram of the relationship between *Arabidopsis thaliana* PIN1–8. Numbers denote bootstrap values of 500 trials. PIN8 is in a clade with the canonical PIN3, PIN4, and PIN7, unlike other non-canonical PINs (PIN5, PIN6). **B)** Figure time course of IAA export by PIN8 from oocytes. Relative IAA content of oocytes expressing PIN8 in the presence (■□) or absence (●) of 10 μM NPA internally determined at the time indicated after substrate injection. Initial internal IAA concentration was 1 μM. n = 10 oocytes at each time point. Data points are mean ± SE. **C)** Raw current traces from SSM-electrophysiology for the PIN8 WT proteoliposomes. **D)** Relative inhibition of the peak binding current induced by 100 μM NPA in the presence of the indicated IAA concentration in non-activating as well as activating buffer. Half-maximal inhibition 39.9 ± 14.9 μM (mean ± SE, n = 3) corresponds to apparent $K_d$(IAA). **E)** Peak currents elicited by 500 μM IAA at the pH indicated (n = 3). Bars are mean ± SE. The points represent individual measurements. **F)** Substrate specificity of PIN8 measured at pH 7.4. Peak currents elicited by IAA (●) or a range of putative substrates tested at 100 μM (▼).

Synthetic auxins: 5-fluoro-IAA, 2,4-D, NAA, TIBA, 1-NOA, 4-CPA, CVX. Endogenous auxins: PAA, IAA-Ala, IBA, Methyl-IAA. Others: Serotonin, Zeatin (a cytokinin), BA (benzoic acid). Chemical structures at pH 7.4 are shown. Current response of substrates indicated with asterisks differed significantly from IAA, indicating that they are likely not substrates for the transporter, but we note that different chemical molecules have different electrostatic potentials and this can also have an influence on the observed current (5-Fluoro IAA p = 0.011; 2,4-D p = 0.272; NAA p = 0.999; TIBA p = 0.989; 1-NOA p = 0.539; PAA p = 0.0007, 4-CPA p < 0.0001; CVX p < 0.0001; IAA-Ala p < 0.0001; IBA p < 0.0001; Methyl-IAA p < 0.0001; Serotonin p < 0.0001; Zeatin p < 0.0001; BA p < 0.0001; IAA+NPPB p < 0.0001; IAA+NPA p < 0.0001). Substrates shown in dark grey are uncharged. Two inhibitors were tested in the presence of 100 μM IAA. Bars are mean ± SE; The data points represent individual measurements. (n = 8: IAA; n = 5: 5-Fluoro IAA, 2,4-D, NAA, TIBA, 4-CPA, IAA-Ala, IBA, Serotonin, Zeatin, BA; n = 3: 1-NOA, PAA, CVX, Methyl-IAA, IAA+NPPB, IAA+NPA).

## Cryo-EM Structure of apo-PIN8 in peptidisc

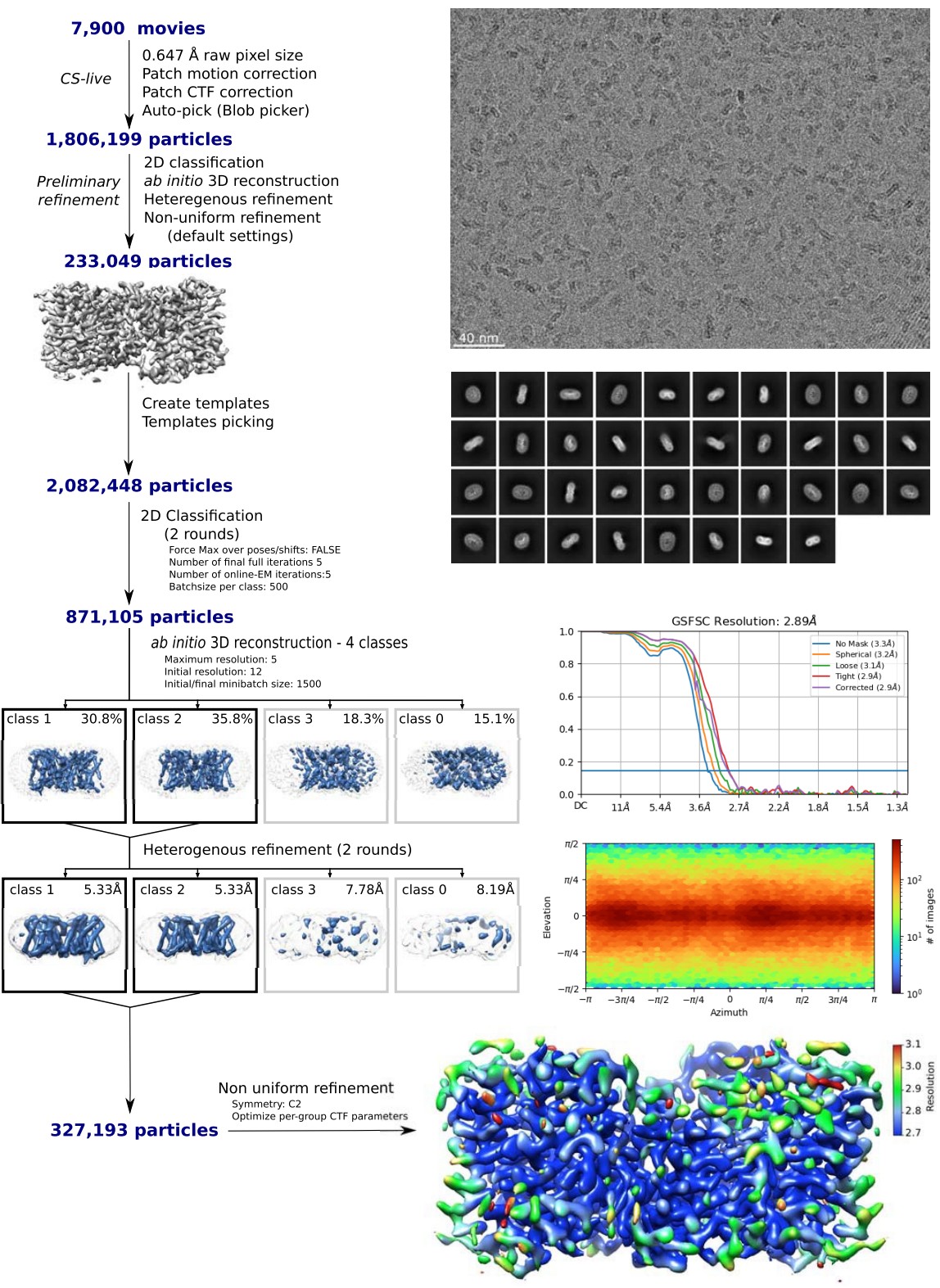

**Extended Data Fig. 3 | Image processing and reconstruction for apo-PIN8.** Workflow of image processing and 3D reconstruction in cryoSPARC, including a motion corrected micrograph from Titan Krios microscope using a K3 detector, 2D classes and sharpened density map from the final non-linear refinement colored by local resolution. Corrected curve of the global Fourier shell correlation (FSC) indicates 2.89 Å based on the 0.143 gold-standard criterion. The cryo-EM experiment with this sample was repeated 7 times with data collection 1 time.

## Cryo-EM Structure of IAA-PIN8 in peptidisc

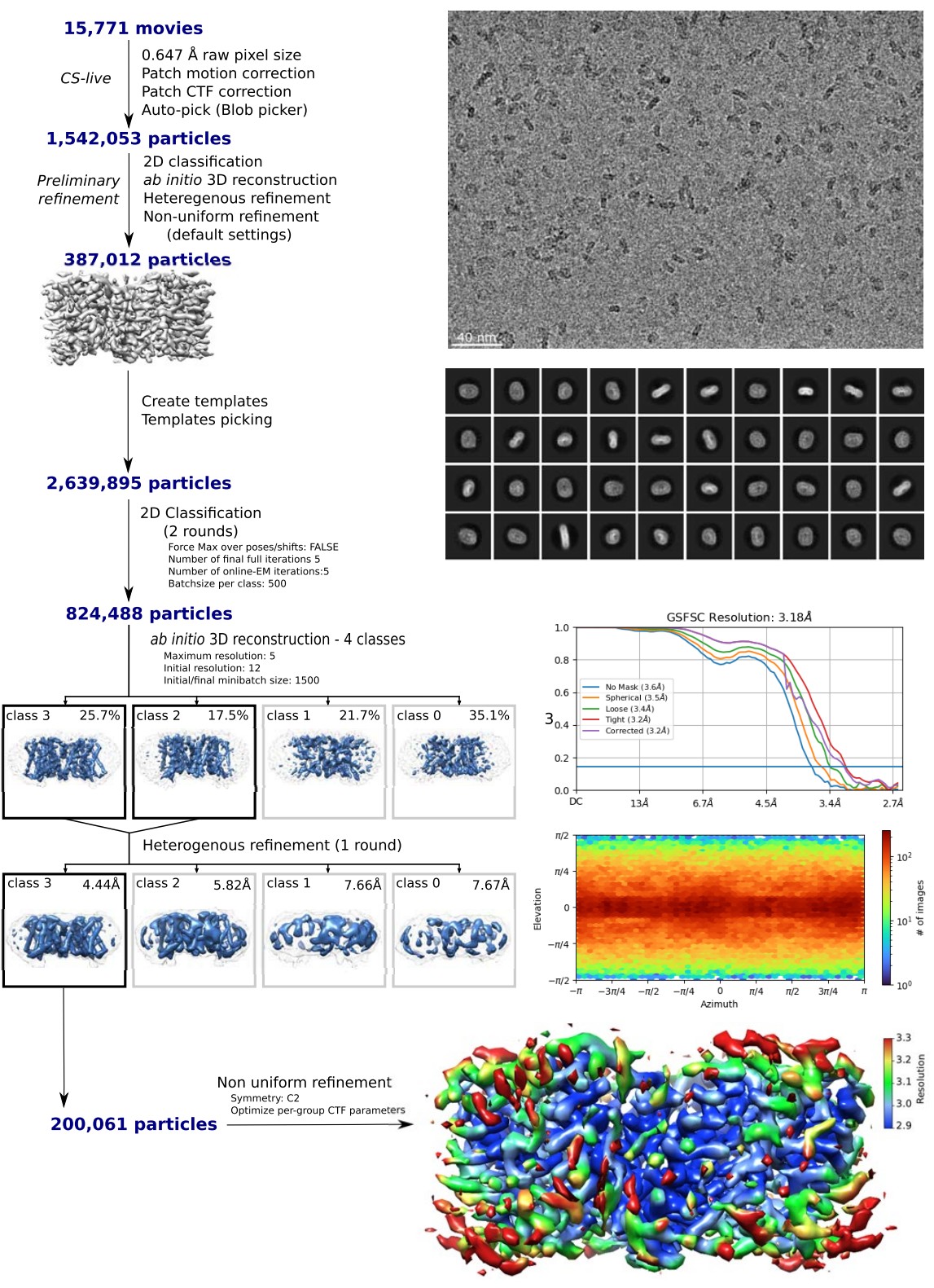

**Extended Data Fig. 4 | Image processing and reconstruction for IAA-PIN8.** Workflow of image processing and 3D reconstruction in cryoSPARC, including a motion corrected micrograph from Titan Krios microscope using a K3 detector, 2D classes and sharpened density map from the final non-linear refinement colored by local resolution. Corrected curve of the global FSC indicates 3.18 Å based on the 0.143 gold-standard criterion. The cryo-EM experiment with this sample was repeated 6 times with data collection 1 time.

## Cryo-EM Structure of NPA-PIN8 in peptidisc

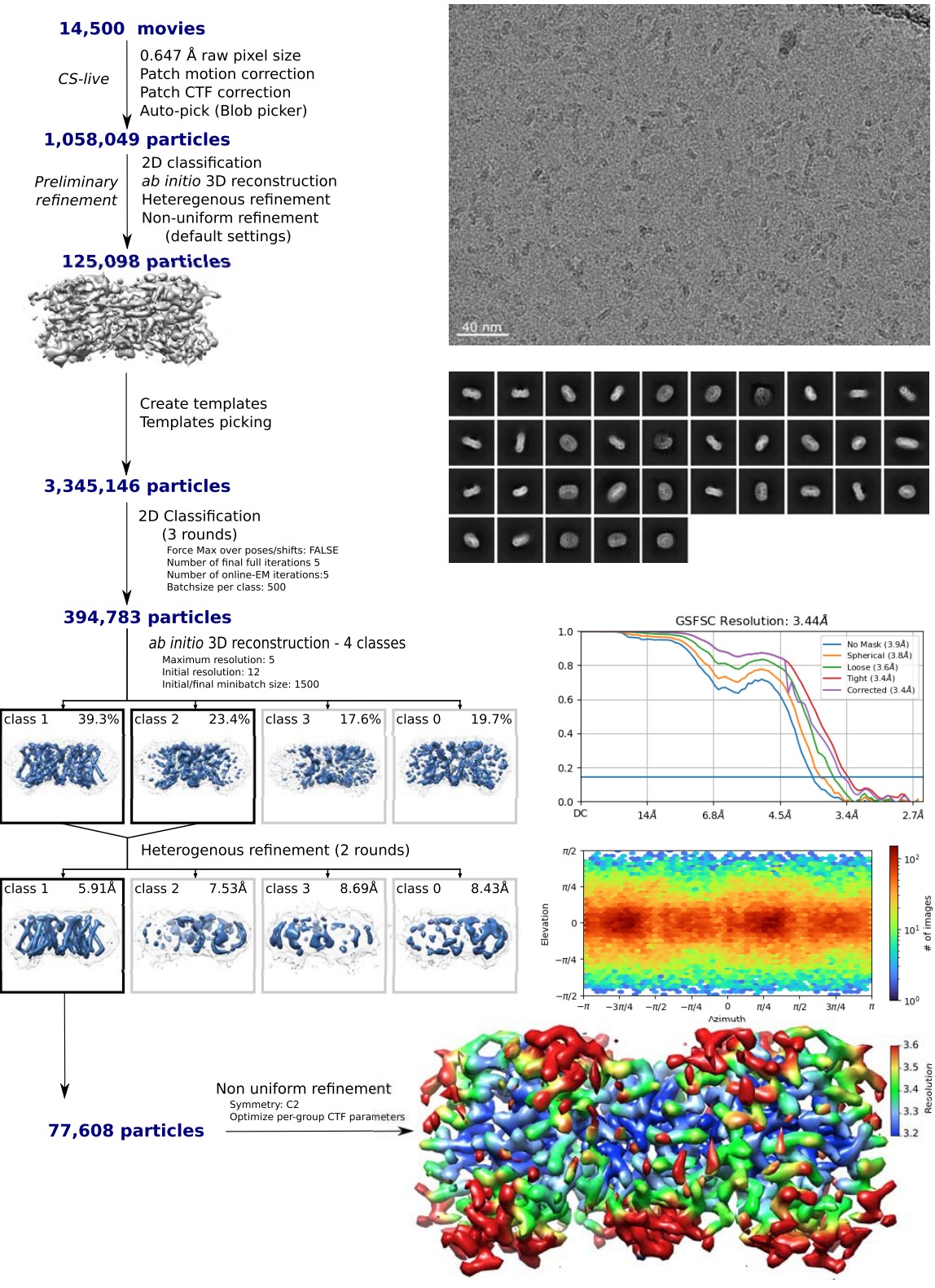

**Extended Data Fig. 5 | Image processing and reconstruction for NPA-PIN8.** Workflow of image processing and 3D reconstruction in cryoSPARC, including a motion corrected micrograph from Titan Krios microscope using a K3 detector, 2D classes and sharpened density map from the final non-linear refinement colored by local resolution. Corrected curve of the global FSC indicates 3.44 Å based on the 0.143 gold-standard criterion. The cryo-EM experiment with this sample was repeated 2 times with data collection 1 time.

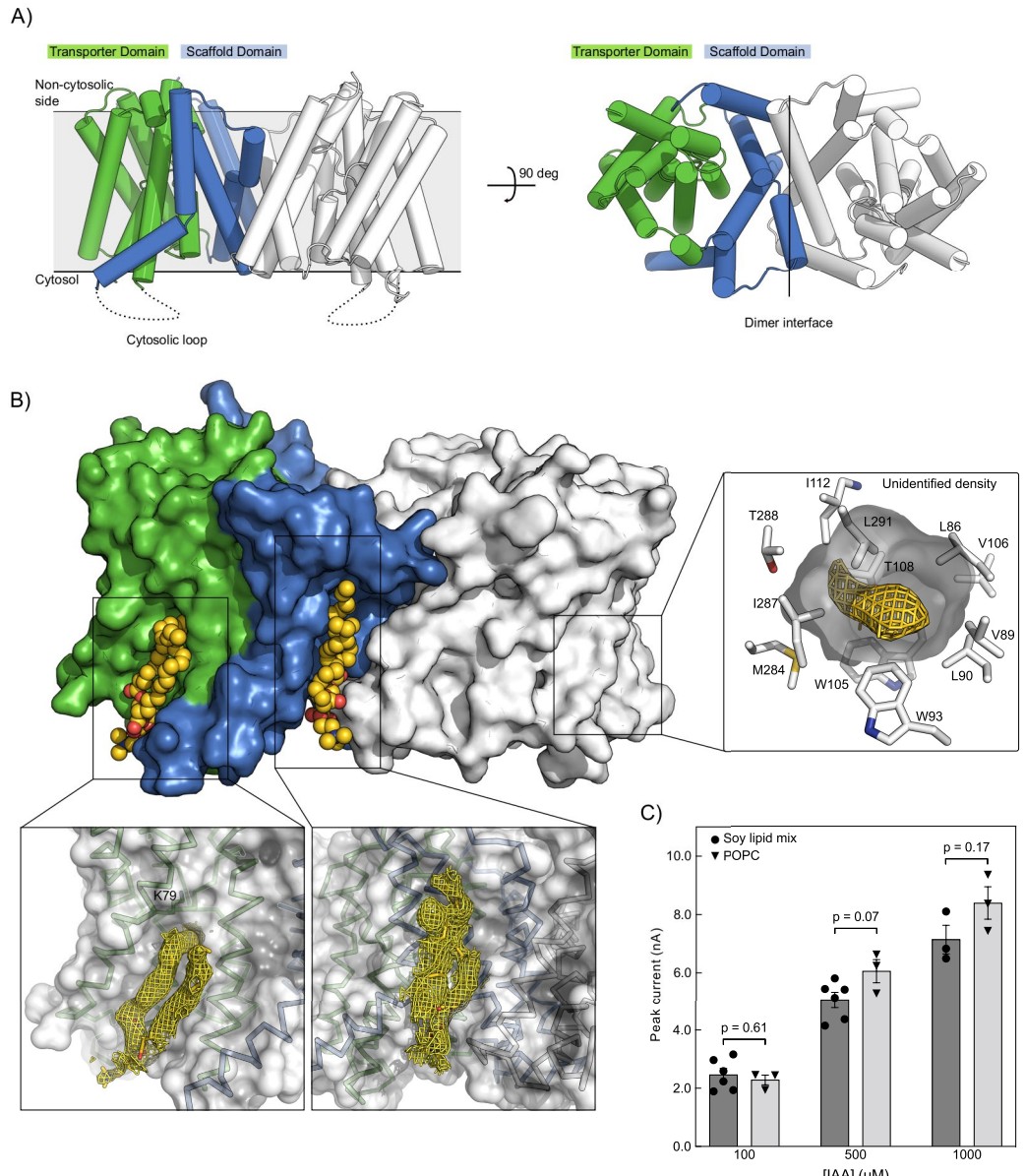

**Extended Data Fig. 6 | Domains and lipids in PIN8. A)** Overview of the transporter (green) and scaffold (blue) domain in the monomer of PIN8. **B)** Position of lipid modeled as phosphatidylcholine in PIN8. One lipid is located in the groove between the two monomers, the other is located at the transporter domain with one aliphatic chain sticking into a cavity of the protein next to the support site. This links the support site to the lipid environment. An unidentified density was found in maps in a cavity towards to the cytosolic side. Mutating T288A did not affect activity (Fig. 2d and Extended Data Fig. 9a).

**C)** Peak currents elicited by the indicated IAA concentrations in liposomes consisting of soy lipid mix (●) or 1-palmitoyl-2-oleoyl-sn-glycero-3-phosphocholine (POPC) (▼). The current response did not differ between the liposomes at any concentration (two-sided unpaired t-test, 100 µM IAA $p = 0.61$, 500 µM IAA $p = 0.07$, 1000 µM $p = 0.17$). This supports that PIN8 is not dependent on specific lipids for activity. Bars are mean ± SE. Data points are independent experiments. (n = 6: soy lipid mix 100 µM and 500 µM; n = 3 all other conditions).

Auxin transporter PIN8 (open outside, PBD: 7QPA)

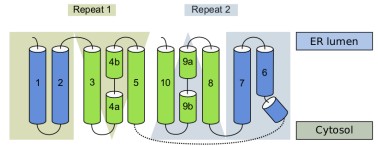

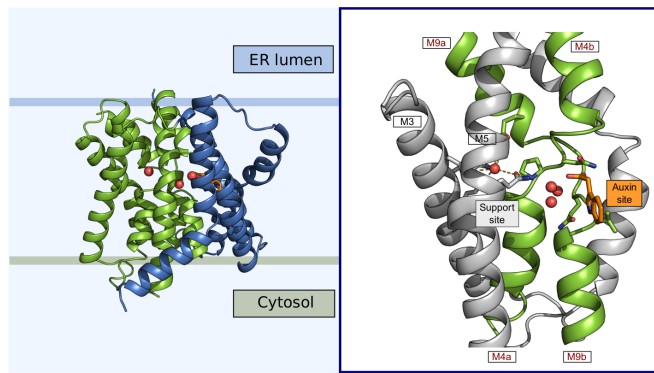

Bicarbonate/sodium symporter SBTA (open inside, PDB: 7EGL)

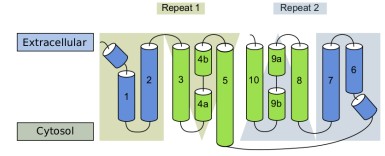

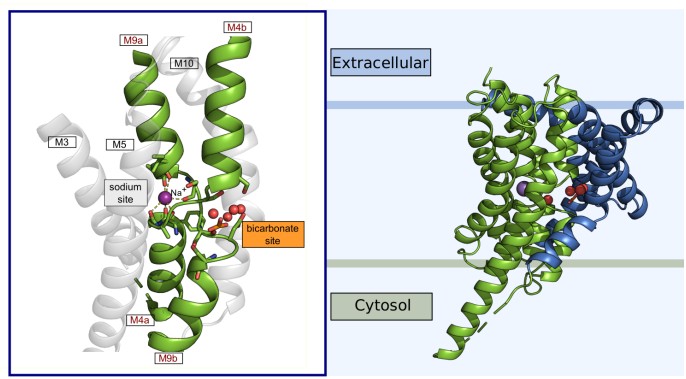

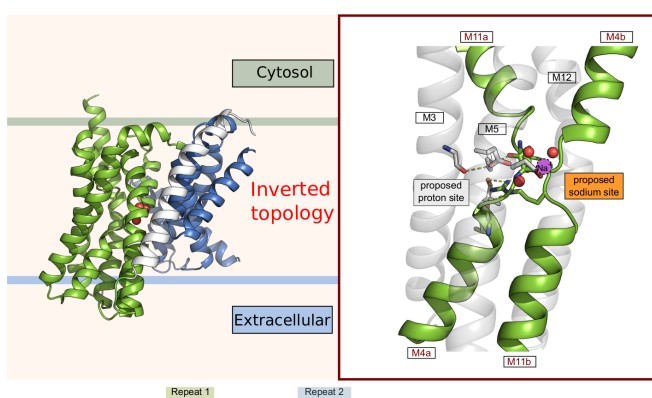

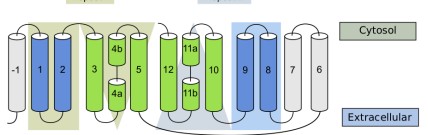

Na/H antiporter NAPA (open outside, PDB: 5BZ3)

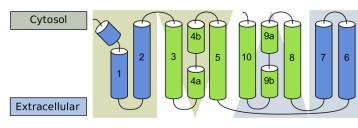

Bile acid/sodium symporter ASBT (open inside, PDB: 3ZUY)

**Extended Data Fig. 7 | Topology of transporters with a crossover elevator mechanism.** Shown are the topologies of auxin transporters (PIN), bicarbonate/sodium symporters (SBTA), Na/H antiporters (NAPA) and bile acid/sodium symporters (ASBT). NAPA and ASBT display an inverted topology compared to PINs and SBTA. All four families have a crossover with a substrate binding site to one side and a putative support site to the other side.

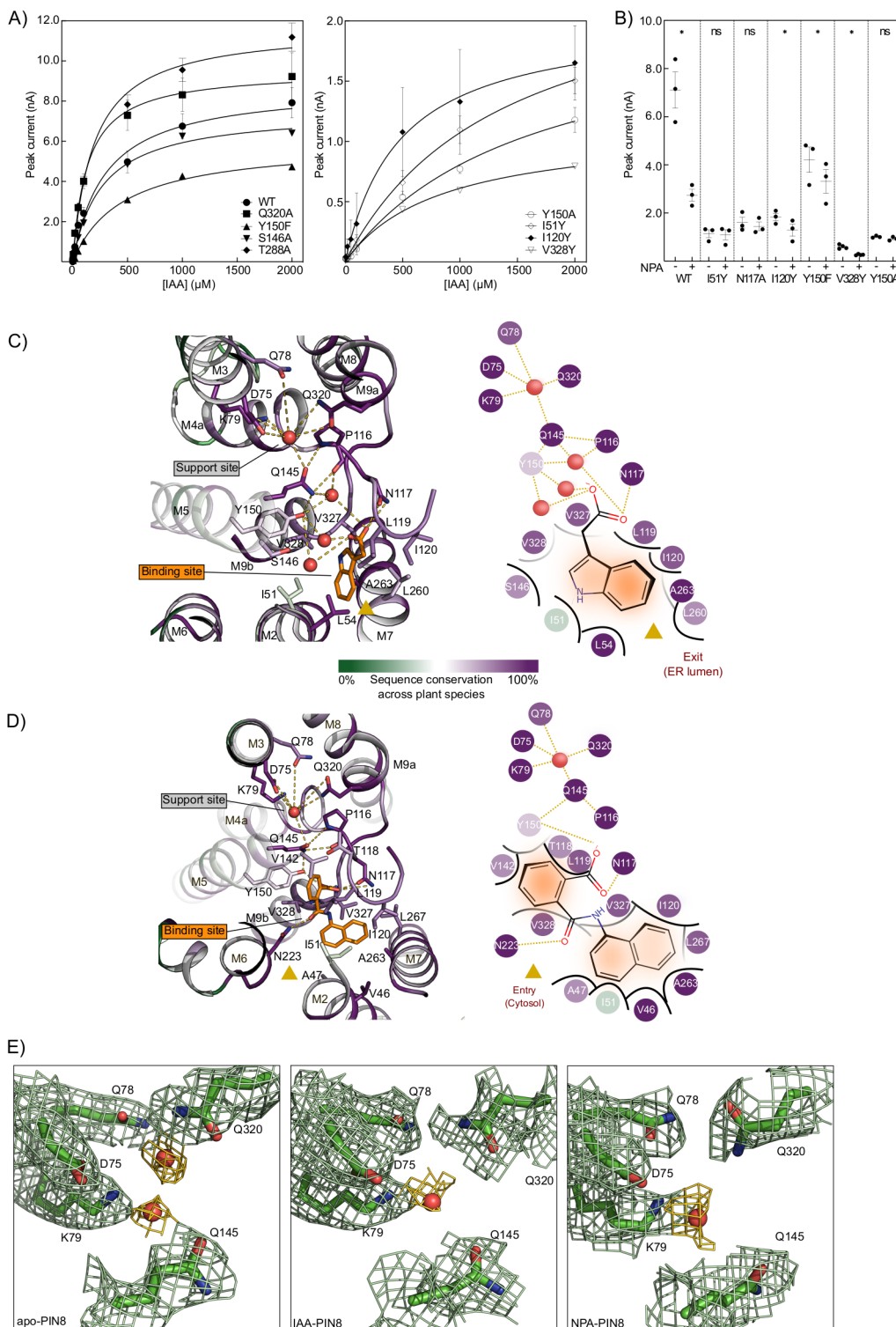

**Extended Data Fig. 8 | Details of mutants and support site. A)** Transport current using SSM-electrophysiology on PIN8 mutants in proteoliposomes. Transport can be described by Michaelis-Menten kinetics. Data points are mean or mean ± SE (n > 2)(WT n = 4 different liposome preparations, for mutants n = 5 (T288A), n = 4 (Q320A), n = 3 (I51Y, I120Y, Y150A), n = 2 (Y150F, S146A, V328Y). **B)** Sensitivity of WT and selected mutants to NPA inhibition. Peak current response to 2 mM IAA or 2 mM IAA and 20 μM NPA presented in non-activating as well as activating buffer. Asterisks indicate significant differences between groups (two-sided paired t-test, WT p = 0.0131, I51Y p = 0.48, N117A p = 0.07, I120Y p = 0.03, Y150F p = 0.02, V328Y p = 0.01, Y150A p = 0.22). Data points are mean ± SE; data points are individual experiments

(n = 4 (V328Y), n = 3 (all other mutants and WT)). **C)** View from the non-cytosolic side of the side chains interacting with IAA and forming the support site. Residues are colored by sequence conservation using ConSurf. 318 unique sequences from plants with sequence identity of 35–95% to AtPIN8 were identified, sorted by E-value and 150 selected at equal intervals for the alignment. **D)** View from the non-cytosolic side of the side chains interacting with NPA and forming the support site. Residues are colored by sequence conservation using ConSurf. **E)** Map density for the peaks found in the support site modeled as water. In the case of apo-PIN two peaks could be modeled as water with one having stronger density than the other.

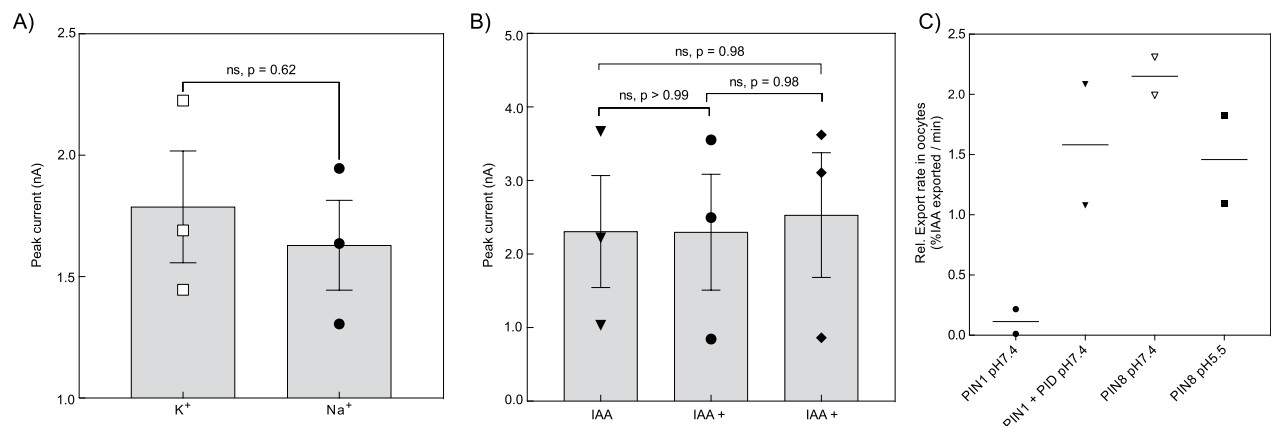

**Extended Data Fig. 9 | Activity assays support PIN8 is independent of ions, pH and lipids. A)** Peak currents elicited by 100 μM IAA in Na⁺-free K⁺ buffer or K⁺-free Na⁺ buffer. The current response was independent of the principal cation. Bars are mean ± SE; n = 3. The points represent individual measurements. Means were compared by a two-tailed unpaired t-test (p = 0.62). **B)** Peak currents elicited by 100 μM IAA, and with proton-motive force decouplers CCCP and DNP present. The current responses were similar in all cases. Bars are mean ± SE; n = 3. The points represent individual measurements. No difference (p = 0.97) between groups was found by one-way ANOVA multiple comparisons, followed by Tukey's Post Hoc test (IAA vs. IAA+CCCP p > 0.99, IAA vs. IAA+DNP p = 0.98, IAA+CCCP vs. IAA+DNP p = 0.98). **C)** Oocyte export assay using ³H-IAA and PIN1 plus kinase PID as a control. PIN8 transport rate is unchanged at two different external pH values; n = 2. Data points are biologically independent experiments. The mean is indicated.

# Extended Data Table 1 | Statistics for cryo-EM data collection, model refinement and validation

**Cryo-EM data collection, refinement and validation statistics**

|  | apo-PIN8 peptidisc | IAA-PIN8 peptidisc | NPA-PIN8 peptidisc | apo-PIN8 LM-NG |
|---|---|---|---|---|
| **Data Collection and processing** | | | | |
| Magnification | 130,000 | 130,000 | 130,000 | 130,000 |
| Voltage (kV) | 300 | 300 | 300 | 300 |
| Electron exposure (e$^-$/Å$^2$) | 59.100 | 60.122 | 59.379 | 60.000 |
| Defocus range (μm) | 0.5-2.5 | 0.4-2.6 | 0.5-2.5 | 0.5-2.5 |
| Pixel size (Å) | 0.647 | 0.647 | 0.647 | 0.653 |
| Symmetry imposed | C2 | C2 | C2 | C2 |
| Collected micrographs (no.) | 7,900 | 15,771 | 14,500 | 7,808 |
| Initial particle images (no.) | 2,082,448 | 2,639,895 | 3,345,146 | 973,540 |
| Final  particle images (no.) | 327,193 | 200,061 | 77,608 | 74,743 |
| Map resolution (Å)* | 2.9 | 3.2 | 3.4 | 3.3 |
| **Refinement** | | | | |
| Initial model used (PDB code) | RosettaFold model | 7QP9 | 7QP9 | |
| Map sharpening B factor (Å$^2$) | -126.9 | -139.8 | -128.3 | |
| Model composition | | | | |
| non-hydrogen atoms | 5,421 | 5,276 | 5,292 | |
| Protein residues | 654 | 654 | 654 | |
| Ligands | DLP: 4 ** | DLP: 2 , IAC: 2 | DLP: 2 , E7O: 2 | |
| Waters | 127 | 64 | 62 | |
| R.m.s. deviations | | | | |
| Bond lenghts (Å) | 0.003 | 0.003 | 0.002 | |
| Bond angles (deg) | 0.486 | 0.554 | 0.513 | |
| Validation | | | | |
| MolProbity score | 1.57 | 1.50 | 1.40 | |
| Clashscore | 6.17 | 8.62 | 5.65 | |
| Poor rotamers (%) | 0 | 0 | 0 | |
| Rama-Z score (Whole/Helix/Loop) | 2.00 / 1.94 / -1.49 | 1.43 / 1.66 / -1.71 | 1.61 / 1.58 / -1.06 | |
| CaBLAM score (Outliers/Disfavored/Cα ) | 0.78 / 3.45 / 0.00 | 0.78 / 5.17 / 0.31 | 0.78 / 4.86 / 0.16 | |
| Ramachandran Plot | | | | |
| Favored (%) | 96.44 | 97.83 | 97.52 | |
| Allowed (%) | 3.56 | 2.17 | 2.48 | |
| Disallowed (%) | 0.00 | 0.00 | 0.00 | |
| Deposited model (PDB id) | 7QP9 | 7QPA | 7QPC | |
| Deposited map (EMDB id) | EMD-14115 | EMD-14116 | EMD-14117 | EMD-14118 |

* Gold standard FSC with threshold of 0.143
** DLP: 1,2-Dilinoleoyl-SN-Glycero-3-Phosphocholine

# Reporting Summary

## Statistics

For all statistical analyses, confirm that the following items are present in the figure legend, table legend, main text, or Methods section.

| n/a | Confirmed | |
|---|---|---|
| ☐ | ☒ | The exact sample size (*n*) for each experimental group/condition, given as a discrete number and unit of measurement |
| ☐ | ☒ | A statement on whether measurements were taken from distinct samples or whether the same sample was measured repeatedly |
| ☐ | ☒ | The statistical test(s) used AND whether they are one- or two-sided<br>*Only common tests should be described solely by name; describe more complex techniques in the Methods section.* |
| ☒ | ☐ | A description of all covariates tested |
| ☐ | ☒ | A description of any assumptions or corrections, such as tests of normality and adjustment for multiple comparisons |
| ☐ | ☒ | A full description of the statistical parameters including central tendency (e.g. means) or other basic estimates (e.g. regression coefficient) AND variation (e.g. standard deviation) or associated estimates of uncertainty (e.g. confidence intervals) |
| ☐ | ☒ | For null hypothesis testing, the test statistic (e.g. *F*, *t*, *r*) with confidence intervals, effect sizes, degrees of freedom and *P* value noted<br>*Give P values as exact values whenever suitable.* |
| ☒ | ☐ | For Bayesian analysis, information on the choice of priors and Markov chain Monte Carlo settings |
| ☒ | ☐ | For hierarchical and complex designs, identification of the appropriate level for tests and full reporting of outcomes |
| ☒ | ☐ | Estimates of effect sizes (e.g. Cohen's *d*, Pearson's *r*), indicating how they were calculated |

*Our web collection on statistics for biologists contains articles on many of the points above.*

## Software and code

Policy information about availability of computer code

| Data collection | EPU 2.11.1.11 |
|---|---|
| Data analysis | CryoSPARC 3.2.0, Chimera 1.14, Phenix 1.19.2, ALINE 1.0.025, Molprobity 4.2, Coot 0.9.4, PyMOL v1.5.0.4, Prism 9.3.1, SURFE2R Control v1.6.0.1, NAMDINATOR v2.0, PROMALS3D (no version), RoseTTAFold (no version),  DALI (no version), NGPhylogeny.fr (no version) |

For manuscripts utilizing custom algorithms or software that are central to the research but not yet described in published literature, software must be made available to editors and reviewers. We strongly encourage code deposition in a community repository (e.g. GitHub). See the Nature Portfolio guidelines for submitting code & software for further information.

## Data

Policy information about availability of data

All manuscripts must include a data availability statement. This statement should provide the following information, where applicable:

- Accession codes, unique identifiers, or web links for publicly available datasets
- A description of any restrictions on data availability
- For clinical datasets or third party data, please ensure that the statement adheres to our policy

Atomic models have been deposited in the Protein Data Bank (PDB) and Cryo-EM maps have been deposited in the Electron Microscopy Data Bank (EMDB). Apo outward state in peptidisc: PDB 7QP9 and EMDB EMD-14115. IAA-bound outward state: PDB 7QPA and EMDB EMD-14116. NPA-bound inward state: PDB 7QPC and EMDB EMD-14117. Apo outward state in detergent: EMDB EMD-14118.
All protein sequences used in this study are publicly available at Uniprot (https://www.uniprot.org/) with the following accession codes. AtPIN1: Q9C6B8, AtPIN2: Q9LU77, AtPIN3: Q9S7Z8, AtPIN4: Q8RWZ6, AtPIN5: Q9FFD0, AtPIN6: Q9SQH6, AtPIN7: Q940Y5, AtPIN8: Q9LFP6.
Source data for biochemical assays in Figures 1 and 2 and Extended data figures 7, 9 and 10 are provided as excel files as part of the supplementary information.

# Field-specific reporting

Please select the one below that is the best fit for your research. If you are not sure, read the appropriate sections before making your selection.

☒ Life sciences ☐ Behavioural & social sciences ☐ Ecological, evolutionary & environmental sciences

For a reference copy of the document with all sections, see nature.com/documents/nr-reporting-summary-flat.pdf

# Life sciences study design

All studies must disclose on these points even when the disclosure is negative.

| | |
|---|---|
| Sample size | Both oocyte and biochemical assays were typically performed at least in triplicate (n=3) to ascertain accurate values for data shown. Statistical methods were not used to determine sample, but were used to calculate standard deviation or standard error as noted in figure legends. |
| Data exclusions | No data was excluded. |
| Replication | All oocyte flux assays except pin8 npa time course were repeated at least once and the data were reproducible. SURFER assays were repeated at least once and the data were reproducible. |
| Randomization | Samples were not randomized for biochemical assays as this is not applicable. |
| Blinding | Blinding was not performed for biochemical assays as this is not applicable. |

# Behavioural & social sciences study design

All studies must disclose on these points even when the disclosure is negative.

| | |
|---|---|
| Study description | *Briefly describe the study type including whether data are quantitative, qualitative, or mixed-methods (e.g. qualitative cross-sectional, quantitative experimental, mixed-methods case study).* |
| Research sample | *State the research sample (e.g. Harvard university undergraduates, villagers in rural India) and provide relevant demographic information (e.g. age, sex) and indicate whether the sample is representative. Provide a rationale for the study sample chosen. For studies involving existing datasets, please describe the dataset and source.* |
| Sampling strategy | *Describe the sampling procedure (e.g. random, snowball, stratified, convenience). Describe the statistical methods that were used to predetermine sample size OR if no sample-size calculation was performed, describe how sample sizes were chosen and provide a rationale for why these sample sizes are sufficient. For qualitative data, please indicate whether data saturation was considered, and what criteria were used to decide that no further sampling was needed.* |
| Data collection | *Provide details about the data collection procedure, including the instruments or devices used to record the data (e.g. pen and paper, computer, eye tracker, video or audio equipment) whether anyone was present besides the participant(s) and the researcher, and whether the researcher was blind to experimental condition and/or the study hypothesis during data collection.* |
| Timing | *Indicate the start and stop dates of data collection. If there is a gap between collection periods, state the dates for each sample cohort.* |
| Data exclusions | *If no data were excluded from the analyses, state so OR if data were excluded, provide the exact number of exclusions and the rationale behind them, indicating whether exclusion criteria were pre-established.* |
| Non-participation | *State how many participants dropped out/declined participation and the reason(s) given OR provide response rate OR state that no participants dropped out/declined participation.* |
| Randomization | *If participants were not allocated into experimental groups, state so OR describe how participants were allocated to groups, and if allocation was not random, describe how covariates were controlled.* |

# Ecological, evolutionary & environmental sciences study design

All studies must disclose on these points even when the disclosure is negative.

| | |
|---|---|
| Study description | *Briefly describe the study. For quantitative data include treatment factors and interactions, design structure (e.g. factorial, nested, hierarchical), nature and number of experimental units and replicates.* |
| Research sample | *Describe the research sample (e.g. a group of tagged Passer domesticus, all Stenocereus thurberi within Organ Pipe Cactus National Monument), and provide a rationale for the sample choice. When relevant, describe the organism taxa, source, sex, age range and* |

| | |
|---|---|
| | *any manipulations. State what population the sample is meant to represent when applicable. For studies involving existing datasets, describe the data and its source.* |
| Sampling strategy | *Note the sampling procedure. Describe the statistical methods that were used to predetermine sample size OR if no sample-size calculation was performed, describe how sample sizes were chosen and provide a rationale for why these sample sizes are sufficient.* |
| Data collection | *Describe the data collection procedure, including who recorded the data and how.* |
| Timing and spatial scale | *Indicate the start and stop dates of data collection, noting the frequency and periodicity of sampling and providing a rationale for these choices. If there is a gap between collection periods, state the dates for each sample cohort. Specify the spatial scale from which the data are taken* |
| Data exclusions | *If no data were excluded from the analyses, state so OR if data were excluded, describe the exclusions and the rationale behind them, indicating whether exclusion criteria were pre-established.* |
| Reproducibility | *Describe the measures taken to verify the reproducibility of experimental findings. For each experiment, note whether any attempts to repeat the experiment failed OR state that all attempts to repeat the experiment were successful.* |
| Randomization | *Describe how samples/organisms/participants were allocated into groups. If allocation was not random, describe how covariates were controlled. If this is not relevant to your study, explain why.* |
| Blinding | *Describe the extent of blinding used during data acquisition and analysis. If blinding was not possible, describe why OR explain why blinding was not relevant to your study.* |

Did the study involve field work?  ☐ Yes  ☐ No

## Field work, collection and transport

| | |
|---|---|
| Field conditions | *Describe the study conditions for field work, providing relevant parameters (e.g. temperature, rainfall).* |
| Location | *State the location of the sampling or experiment, providing relevant parameters (e.g. latitude and longitude, elevation, water depth).* |
| Access & import/export | *Describe the efforts you have made to access habitats and to collect and import/export your samples in a responsible manner and in compliance with local, national and international laws, noting any permits that were obtained (give the name of the issuing authority, the date of issue, and any identifying information).* |
| Disturbance | *Describe any disturbance caused by the study and how it was minimized.* |

# Reporting for specific materials, systems and methods

We require information from authors about some types of materials, experimental systems and methods used in many studies. Here, indicate whether each material, system or method listed is relevant to your study. If you are not sure if a list item applies to your research, read the appropriate section before selecting a response.

### Materials & experimental systems

| n/a | Involved in the study |
|---|---|
| ☒ | ☐ Antibodies |
| ☒ | ☐ Eukaryotic cell lines |
| ☒ | ☐ Palaeontology and archaeology |
| ☒ | ☐ Animals and other organisms |
| ☒ | ☐ Human research participants |
| ☒ | ☐ Clinical data |
| ☒ | ☐ Dual use research of concern |

### Methods

| n/a | Involved in the study |
|---|---|
| ☒ | ☐ ChIP-seq |
| ☒ | ☐ Flow cytometry |
| ☒ | ☐ MRI-based neuroimaging |

## Antibodies

| | |
|---|---|
| Antibodies used | *Describe all antibodies used in the study; as applicable, provide supplier name, catalog number, clone name, and lot number.* |
| Validation | *Describe the validation of each primary antibody for the species and application, noting any validation statements on the manufacturer's website, relevant citations, antibody profiles in online databases, or data provided in the manuscript.* |

## Eukaryotic cell lines

Policy information about cell lines

| | |
|---|---|
| Cell line source(s) | *State the source of each cell line used.* |

| Authentication | *Describe the authentication procedures for each cell line used OR declare that none of the cell lines used were authenticated.* |
|---|---|
| Mycoplasma contamination | *Confirm that all cell lines tested negative for mycoplasma contamination OR describe the results of the testing for mycoplasma contamination OR declare that the cell lines were not tested for mycoplasma contamination.* |
| Commonly misidentified lines (See ICLAC register) | *Name any commonly misidentified cell lines used in the study and provide a rationale for their use.* |

# Palaeontology and Archaeology

| Specimen provenance | *Provide provenance information for specimens and describe permits that were obtained for the work (including the name of the issuing authority, the date of issue, and any identifying information). Permits should encompass collection and, where applicable, export.* |
|---|---|
| Specimen deposition | *Indicate where the specimens have been deposited to permit free access by other researchers.* |
| Dating methods | *If new dates are provided, describe how they were obtained (e.g. collection, storage, sample pretreatment and measurement), where they were obtained (i.e. lab name), the calibration program and the protocol for quality assurance OR state that no new dates are provided.* |

☐ Tick this box to confirm that the raw and calibrated dates are available in the paper or in Supplementary Information.

| Ethics oversight | *Identify the organization(s) that approved or provided guidance on the study protocol, OR state that no ethical approval or guidance was required and explain why not.* |
|---|---|

Note that full information on the approval of the study protocol must also be provided in the manuscript.

# Animals and other organisms

Policy information about studies involving animals; ARRIVE guidelines recommended for reporting animal research

| Laboratory animals | *For laboratory animals, report species, strain, sex and age OR state that the study did not involve laboratory animals.* |
|---|---|
| Wild animals | *Provide details on animals observed in or captured in the field; report species, sex and age where possible. Describe how animals were caught and transported and what happened to captive animals after the study (if killed, explain why and describe method; if released, say where and when) OR state that the study did not involve wild animals.* |
| Field-collected samples | *For laboratory work with field-collected samples, describe all relevant parameters such as housing, maintenance, temperature, photoperiod and end-of-experiment protocol OR state that the study did not involve samples collected from the field.* |
| Ethics oversight | *Identify the organization(s) that approved or provided guidance on the study protocol, OR state that no ethical approval or guidance was required and explain why not.* |

Note that full information on the approval of the study protocol must also be provided in the manuscript.

# Human research participants

Policy information about studies involving human research participants

| Population characteristics | *Describe the covariate-relevant population characteristics of the human research participants (e.g. age, gender, genotypic information, past and current diagnosis and treatment categories). If you filled out the behavioural & social sciences study design questions and have nothing to add here, write "See above."* |
|---|---|
| Recruitment | *Describe how participants were recruited. Outline any potential self-selection bias or other biases that may be present and how these are likely to impact results.* |
| Ethics oversight | *Identify the organization(s) that approved the study protocol.* |

Note that full information on the approval of the study protocol must also be provided in the manuscript.

# Clinical data

Policy information about clinical studies

All manuscripts should comply with the ICMJE guidelines for publication of clinical research and a completed CONSORT checklist must be included with all submissions.

| Clinical trial registration | *Provide the trial registration number from ClinicalTrials.gov or an equivalent agency.* |
|---|---|
| Study protocol | *Note where the full trial protocol can be accessed OR if not available, explain why.* |
| Data collection | *Describe the settings and locales of data collection, noting the time periods of recruitment and data collection.* |
| Outcomes | *Describe how you pre-defined primary and secondary outcome measures and how you assessed these measures.* |

# Dual use research of concern

Policy information about dual use research of concern

## Hazards

Could the accidental, deliberate or reckless misuse of agents or technologies generated in the work, or the application of information presented in the manuscript, pose a threat to:

No | Yes

- ☐ | ☐ Public health
- ☐ | ☐ National security
- ☐ | ☐ Crops and/or livestock
- ☐ | ☐ Ecosystems
- ☐ | ☐ Any other significant area

## Experiments of concern

Does the work involve any of these experiments of concern:

No | Yes

- ☐ | ☐ Demonstrate how to render a vaccine ineffective
- ☐ | ☐ Confer resistance to therapeutically useful antibiotics or antiviral agents
- ☐ | ☐ Enhance the virulence of a pathogen or render a nonpathogen virulent
- ☐ | ☐ Increase transmissibility of a pathogen
- ☐ | ☐ Alter the host range of a pathogen
- ☐ | ☐ Enable evasion of diagnostic/detection modalities
- ☐ | ☐ Enable the weaponization of a biological agent or toxin
- ☐ | ☐ Any other potentially harmful combination of experiments and agents

# ChIP-seq

## Data deposition

☐ Confirm that both raw and final processed data have been deposited in a public database such as GEO.

☐ Confirm that you have deposited or provided access to graph files (e.g. BED files) for the called peaks.

Data access links
*May remain private before publication.*
| *For "Initial submission" or "Revised version" documents, provide reviewer access links. For your "Final submission" document, provide a link to the deposited data.*

Files in database submission | *Provide a list of all files available in the database submission.*

Genome browser session
(e.g. UCSC) | *Provide a link to an anonymized genome browser session for "Initial submission" and "Revised version" documents only, to enable peer review. Write "no longer applicable" for "Final submission" documents.*

## Methodology

Replicates | *Describe the experimental replicates, specifying number, type and replicate agreement.*

Sequencing depth | *Describe the sequencing depth for each experiment, providing the total number of reads, uniquely mapped reads, length of reads and whether they were paired- or single-end.*

Antibodies | *Describe the antibodies used for the ChIP-seq experiments; as applicable, provide supplier name, catalog number, clone name, and lot number.*

Peak calling parameters | *Specify the command line program and parameters used for read mapping and peak calling, including the ChIP, control and index files used.*

Data quality | *Describe the methods used to ensure data quality in full detail, including how many peaks are at FDR 5% and above 5-fold enrichment.*

Software | *Describe the software used to collect and analyze the ChIP-seq data. For custom code that has been deposited into a community repository, provide accession details.*

# Flow Cytometry

## Plots

Confirm that:

☐ The axis labels state the marker and fluorochrome used (e.g. CD4-FITC).

☐ The axis scales are clearly visible. Include numbers along axes only for bottom left plot of group (a 'group' is an analysis of identical markers).

☐ All plots are contour plots with outliers or pseudocolor plots.

☐ A numerical value for number of cells or percentage (with statistics) is provided.

## Methodology

| | |
|---|---|
| Sample preparation | *Describe the sample preparation, detailing the biological source of the cells and any tissue processing steps used.* |
| Instrument | *Identify the instrument used for data collection, specifying make and model number.* |
| Software | *Describe the software used to collect and analyze the flow cytometry data. For custom code that has been deposited into a community repository, provide accession details.* |
| Cell population abundance | *Describe the abundance of the relevant cell populations within post-sort fractions, providing details on the purity of the samples and how it was determined.* |
| Gating strategy | *Describe the gating strategy used for all relevant experiments, specifying the preliminary FSC/SSC gates of the starting cell population, indicating where boundaries between "positive" and "negative" staining cell populations are defined.* |

☐ Tick this box to confirm that a figure exemplifying the gating strategy is provided in the Supplementary Information.

# Magnetic resonance imaging

## Experimental design

| | |
|---|---|
| Design type | *Indicate task or resting state; event-related or block design.* |
| Design specifications | *Specify the number of blocks, trials or experimental units per session and/or subject, and specify the length of each trial or block (if trials are blocked) and interval between trials.* |
| Behavioral performance measures | *State number and/or type of variables recorded (e.g. correct button press, response time) and what statistics were used to establish that the subjects were performing the task as expected (e.g. mean, range, and/or standard deviation across subjects).* |

## Acquisition

| | |
|---|---|
| Imaging type(s) | *Specify: functional, structural, diffusion, perfusion.* |
| Field strength | *Specify in Tesla* |
| Sequence & imaging parameters | *Specify the pulse sequence type (gradient echo, spin echo, etc.), imaging type (EPI, spiral, etc.), field of view, matrix size, slice thickness, orientation and TE/TR/flip angle.* |
| Area of acquisition | *State whether a whole brain scan was used OR define the area of acquisition, describing how the region was determined.* |

Diffusion MRI     ☐ Used     ☐ Not used

## Preprocessing

| | |
|---|---|
| Preprocessing software | *Provide detail on software version and revision number and on specific parameters (model/functions, brain extraction, segmentation, smoothing kernel size, etc.).* |
| Normalization | *If data were normalized/standardized, describe the approach(es): specify linear or non-linear and define image types used for transformation OR indicate that data were not normalized and explain rationale for lack of normalization.* |
| Normalization template | *Describe the template used for normalization/transformation, specifying subject space or group standardized space (e.g. original Talairach, MNI305, ICBM152) OR indicate that the data were not normalized.* |
| Noise and artifact removal | *Describe your procedure(s) for artifact and structured noise removal, specifying motion parameters, tissue signals and physiological signals (heart rate, respiration).* |

| Volume censoring | *Define your software and/or method and criteria for volume censoring, and state the extent of such censoring.* |
|---|---|

## Statistical modeling & inference

| Model type and settings | *Specify type (mass univariate, multivariate, RSA, predictive, etc.) and describe essential details of the model at the first and second levels (e.g. fixed, random or mixed effects; drift or auto-correlation).* |
|---|---|
| Effect(s) tested | *Define precise effect in terms of the task or stimulus conditions instead of psychological concepts and indicate whether ANOVA or factorial designs were used.* |

Specify type of analysis: ☐ Whole brain ☐ ROI-based ☐ Both

| Statistic type for inference (See Eklund et al. 2016) | *Specify voxel-wise or cluster-wise and report all relevant parameters for cluster-wise methods.* |
|---|---|
| Correction | *Describe the type of correction and how it is obtained for multiple comparisons (e.g. FWE, FDR, permutation or Monte Carlo).* |

## Models & analysis

| n/a | Involved in the study |
|---|---|
| ☐ | ☐ Functional and/or effective connectivity |
| ☐ | ☐ Graph analysis |
| ☐ | ☐ Multivariate modeling or predictive analysis |

| Functional and/or effective connectivity | *Report the measures of dependence used and the model details (e.g. Pearson correlation, partial correlation, mutual information).* |
|---|---|
| Graph analysis | *Report the dependent variable and connectivity measure, specifying weighted graph or binarized graph, subject- or group-level, and the global and/or node summaries used (e.g. clustering coefficient, efficiency, etc.).* |
| Multivariate modeling and predictive analysis | *Specify independent variables, features extraction and dimension reduction, model, training and evaluation metrics.* |

