## [Peer Review File · Nature]

Manuscript Title: Structures and mechanism of the plant PIN-FORMED auxin transporter

Reviewer Comments & Author Rebuttals

Reviewer Reports on the Initial Version:

Referee #1 (Remarks to the Author):

The manuscript by Ung et al. reports three cryo-EM structures of the PIN8 auxin transporter from *A. thaliana*. The three structures represent two outward open states (Apo and IAA bound) and inward open (NPA bound). The structures are determined at a suitable resolution to accurately model in the ligands, observe water molecules and map out the main interactions. The authors use both an oocyte assay (radioactive IAA uptake) and in vitro reconstituted assay employing solid supported membrane (SSM) based electrophysiology.

The work is highly original in that they report the first structure and in vitro functional analysis of a plant PIN-FORMED protein, PIN8. The architecture resembles that for previously determined secondary active transporters including the bile acid sodium symporters, bicarbonate sodium symporters and sodium proton antiporters, although with some interesting differences and functional adaptations, discussed below.

The work is very suitable for publication in Nature in my opinion, and reports fascinating and important new insights into plant transport and IAA regulation. The insights into NPA binding and mechanism of action will also likely spur innovation in the herbicide field.

Nevertheless, the work does suffer from some over interpretation in my opinion, which I would like the authors to address/comment on.

- It appears to me that PIN8 selects for IAA based on shape complementarity within the binding pocket, yet this is not really spelt out in the main text. In ED Fig. 2D they report the results of transport for a panel of auxin compounds, but interestingly not PAA (2-phenylacetic acid)? Wouldn't this nicely show how important the indole ring is for affinity, just as the reduced current for IBA shows that longer ligands are not recognised well? I felt this part of the analysis was too cursory and

could be expanded to draw conclusions between the chemistry of the ligands and their ability to be transported. Minor comment - please add the chemical structures to this ED Figure.

- Line 72. The authors say that NPA is a classical competitive inhibitor, yet don't mention the effect on V_{max} . Indeed, the effect of V_{max} on any aspect of the study (inhibitors/mutants) is strangely missing from the data. Without this parameter, it is not possible to say with certainty this is classical competitive from the kinetic data alone.

- In connection with the point above, the authors do not much discuss the effects of the mutants in Figure 2D. This is very odd as surely this is rather interesting. For example, the authors state that "mutation of Ser146 have little effect on activity". I disagree, the K_m drops by half and displays a higher peak current than WT. I think this is very interesting.. what is changing here, the K_m or the V_{max} ...? Similarly, what happens when you increase the concentration of IAA for the mutants that display higher K_m values? Does the peak current change or not? This could tell you whether the mutants are kinetically restricted, i.e slower. Does the current trace tell you anything about the kinetics? This part of the analysis is very cursory and I feel there is a lot that can be determined from this data which is currently lacking.

- The structure of the NPA bound state is also interesting. I assume this result, combined with the fact that NPA inhibits from the extracellular side of the oocyte, suggests that NPA binds from the apoplast, whereupon PIN8 adopts the inward open state and the system stalls in this position because NPA cannot unbind into the cytoplasm. However, the authors do not really address what holds NPA in the binding site. I feel here, SPA (scintillation proximity assays) could have been employed to more fully understand this mechanism. However, in the absence of a binding assay to probe the importance of the interactions, the authors could investigate what effects certain functional mutations have on the K_i . The Y150F/A mutants for example look rather interesting.

- Line 95. The authors show in ED. Fig. 7C that lipids have no effect on PIN8 but do not explain what the significance of this observation, if any? I think at least some context here would be helpful.

- In connection with the above, the authors also reveal an unexplained density on the cytoplasmic side of the transporter that appeared in all structures. One assumes then that this molecule must come from the yeast expression system. This raises to questions in my mind, which may or may not be troubling..

1. Is this site regulatory in nature? If so, is the molecule something that is present in the yeast cytoplasm that could mimic an endogenous regulator of PIN8 in planta? I assume the authors tried to model in IAA? 2. How confident are the authors that this molecule is not interfering with their kinetic analyses? For example, the authors state that the K_m is much lower than for PIN8 in vivo,

could this be the reason? The site is similar in nature to the binding site, mainly hydrophobic with a single H-bond donor/acceptor (Thr288) - unless I misread the figure? Have the authors mutated residues in this site to assess the effect on transport? T288A for example? It certainly leaves open the possibility that all parameters may not have been fully accounted for in the analysis of the data.

- The last section of the paper is too speculative for me. I very much enjoyed reading the paper, but the authors must admit there is not evidence in this paper to support any mechanistic role for proton binding/movement between D75 and K79. I appreciate the structural similarities to other SLC systems, but this is presented and discussed as if the 'support site' is mechanistically important and validated, which it isn't. It is perfectly conceivable, in my view, that this site has evolved to make PIN8 independent of pH or Na⁺ gradients, i.e. convert an antiporter into a symporter and nothing more. I think the authors should address this in their rebuttal and amend the text and proposed mechanism in Figure 4. Or very clearly say this part of the charge-diffusion mechanism is pure speculation at this point and isn't based on any data presented herein.

Minor comments

- The water molecules in the main figures should be colored something other than red, which is difficult to discriminate from the ligand carboxyl group.

- Drawing the hydrogen bonds would be helpful in the main figure as in the ED figures.

Referee #2 (Remarks to the Author):

Ung et al. report the structures of Arabidopsis PIN8 protein, a member of the auxin efflux carrier family that has been less well-studied than the "canonical" PIN proteins with long hydrophilic loops. However, transport assays have shown that the efflux activity is conserved among the entire family. Therefore, PIN8 is probably a reasonable proxy for PIN protein structure and function. The auxin field has eagerly awaited the first PIN protein structures. The authors have leveraged the use of Cryo-EM on PIN8 protein expressed in yeast to generate structures of the apo-form, as well as IAA-bound and NPA-bound forms of the protein. Through analyzing these structures, their similarities to Bile Acid / Na transporters, through biochemical analysis and transport assays and mutant characterization, the authors present a mechanism for IAA transport by the PIN8 protein, as well as for its inhibition by NPA. This is a truly groundbreaking finding, that will open many new avenues of investigation and helps rationalize prior findings, such as the dimerization of PIN proteins and

binding of NPA to PIN proteins. PIN8 has been studied much less intensively than canonical PIN proteins, and is not exposed to the same environments as canonical PINs. While canonical PINs are exposed to apoplast on the outside and cytosol in the inside, PIN8 is ER-localized, and is exposed to the ER lumen on the “outside”. This has bearing on how well findings can be extrapolated (see point 2 below), but the strong conservation of key residues in the PIN family gives confidence that the mechanisms are also conserved.

While of major importance, there are some points of attention:

1. It is unfortunate that a manuscript of this kind, reporting such an important breakthrough in auxin biology, lacks any form of *in vivo* validation. The authors admittedly developed a very good oocyte-based assay for auxin transport in which mutants are tested, but this is far from the natural context, where ER-localized PIN8 mediates specific functions. Given that there are *pin8* mutants with well-described phenotypes, it would have been relatively straight-forward to complement the *pin8* mutant with versions in which key residues are mutated. I believe this is critical to reach the full impact of this study, and to translate *in vitro* findings to the biological context.
2. As indicated above, PIN8 is an ER-localized protein, and this has consequences for the milieu that the protein faces are exposed to. If the protein expressed in oocytes for functional assays is in fact in the plasma membrane (which would be required to get a functional read-out in a transport assay – but is not shown), the sensitivity to pH, uncouplers, ions, etc, is perhaps not reflecting its normal activity. This needs to be explicitly addressed.
3. The authors explored the specificity of transport mediated by PIN8 in heterologous assays, and find that 2,4-D and NAA are about as effective as IAA in triggering a current in oocytes. This needs explanation, given that 2,4-D is used as a non-transportable (i.e. non-substrate) auxin *in vivo*.
4. The authors identify NPA-interacting residues. Some of these are mutated and tested for activity (Figure 2D), but whether these proteins are still sensitive to NPA is not assayed. This would be important to conclude on the mechanism of NPA action.
5. PIN8 has low affinity to IAA, when contrasted to predicted cellular auxin concentrations. The interpretation is not entirely clear to this reviewer. Is it possible that the polar accumulation of PIN proteins serves the role of generating high local protein concentrations to overcome low affinity for IAA binding? If so, how does this connect to the ER localization of PIN8?

6. It is interesting that PIN8 shows reasonable structural similarity to Bile Acid / Na transporters. Given the low sequence homology, it is perhaps not strange that this was previously undetected. With this knowledge however, it would be interesting to see how well PIN protein structure could be modeled using these templates.

7. Likewise, the option of hetero-dimerization (or oligomerization) of PIN proteins is mentioned. It would be interesting to use modeling/docking to explore further if this is realistic.

Minor issues:

1. The term “morphogen” has a very specific connotation, and there is no convincing or conclusive evidence that auxin acts as a morphogen

2. “Most auxin effects are due to polar auxin transport” sounds strange...it is the actual response that triggers cells to at, so at best, PAT is involved in these, or controlling these.

3. It would be helpful to describe the screening process that led to the selection of PIN8 for detailed investigation.

Referee #3 (Remarks to the Author):

Auxin is a group of phytohormone and has critical roles in virtually all aspects of plant development. IAA is the most common naturally occurring of auxin. PINs are a family of membrane proteins that transport auxin across cells and tissues. Many studies have been directed at the developmental roles of PINs, very limited knowledge is available on their structures. The current manuscript reports biochemical and structural characterization of the short PIN protein Arabidopsis PIN8 (AtPIN8). Electrophysiological data showed that AtPIN8 was constitutively active in transporting auxin. The authors then solved cryo-EM structures of AtPIN8 in apo-, substrate IAA- and the inhibitor NPA-bound AtPIN8 forms. The structures showed that AtPIN8 contains the scaffold and transport domains and forms a dimer. The structures revealed that Apo- and IAA-bound AtPIN8 adopt an outward-open conformation, whereas the NPA-bound AtPIN8 assumes an inward-open conformation. Structural comparison indicated that NPA binding causes rotation of about 20 degrees of the two transport domains of the AtPIN8 dimer. Data from mutagenesis analysis provided further evidence for recognition of IAA and NPA by AtPIN8. DALI search identified several families of secondary active transporters as close structural homologs of AtPIN8. Interestingly, these proteins are believed to function via an elevator type transport mechanism in which the transport domain

undergo striking conformational changes during transport. The manuscript was well written and the structural data are solid. The results presented in the manuscript are significant for understanding the transport mechanism of auxin by AtPINs. However, a number of issues need addressing before it can be accepted by Nature.

1. Based on the cryo-EM the structures in the manuscript, can the data presented in Extended Data Fig. 2D be explained? For example, 2,4-D differs from and 4-CPA only in one chloride atom. However, AtPIN8 displayed a much higher toward to the former than toward the latter.
2. The authors concluded that the negative charge of IAA is sufficient for AtPIN8-mediated transport. This is consistent with the on the chemiosmotic model of auxin transport. However, changes in pH only slightly affected the activity of transporting IAA by AtPIN8 (Extended Data Fig. 2D). Please make comments on this.
3. The IAA-binding affinity of AtPIN8 is extremely low, about 500-fold lower that the physiological concentrations of auxin. The authors have to verify the IAA-binding activity of AtPIN1 using other biophysical methods like ITC.
4. The major issue the reviewer had with the manuscript is about the proposed model on auxin transport by PINs (Fig. 4). It is true that the elevator mechanism has been suggested for some transporters. However, other alternating mechanisms like rocker switch and rocking bundle were also proposed for transporters. Can these two mechanisms be excluded by the data presented in the manuscript? The elevator-like model does not provide an explanation of why IAA is a substrate but NPA is an inhibitor of PINs. Evidence for the elevator mechanism of PIN-mediated auxin transport is that AtPIN8 bears structural similarity to some secondary active transporters. Structural comparison did reveal striking conformation differences between the apo-/IAA- and the NPA-bound forms of AtPIN8. While not described in the manuscript, I assumed that the authors used the transport domain as template for the structure alignment. However, striking conformation differences between the apo-/IAA- and the NPA-bound forms of AtPIN8 similarly exist in the scaffold domain if the transport domain is used as template for structural alignment. To the reviewer, this is more possible and even likely for at least for reasons. First, the major IAA-/NPA-binding site is located at the X-shaped crossover; second, such alignment puts IAA and NPA in a similar site around the crossover. IAA binding to the inward-open (NPA-bound) form of AtPIN8 generates steric clashes with Ile50, triggering conformational changes in the scaffold domain and leading to the outward-open state. In contrast, NPA binding is unable to trigger conformational changes in scaffold domain, blocking AtPIN8 conversion into an outward-open state.
5. AtPIN8 forms a homodimer. But whether it is functionally required remains unclear.

Author Rebuttals to Initial Comments:

Reply to Referees.

(our answer in blue)

Referee #1 (Remarks to the Author):

The manuscript by Ung et al. reports three cryo-EM structures of the PIN8 auxin transporter from *A. thaliana*. The three structures represent two outward open states (Apo and IAA bound) and inward open (NPA bound). The structures are determined at a suitable resolution to accurately model in the ligands, observe water molecules and map out the main interactions. The authors use both an oocyte assay (radioactive IAA uptake) and in vitro reconstituted assay employing solid supported membrane (SSM) based electrophysiology.

The work is highly original in that they report the first structure and in vitro functional analysis of a plant PIN-FORMED protein, PIN8. The architecture resembles that for previously determined secondary active transporters including the bile acid sodium symporters, bicarbonate sodium symporters and sodium proton antiporters, although with some interesting differences and functional adaptations, discussed below.

The work is very suitable for publication in Nature in my opinion, and reports fascinating and important new insights into plant transport and IAA regulation. The insights into NPA binding and mechanism of action will also likely spur innovation in the herbicide field.

We thank the referee for the strong support of our work.

Nevertheless, the work does suffer from some over interpretation in my opinion, which I would like the authors to address/comment on.

In general, we have toned down the more speculative aspects of the manuscript as discussed specifically below.

- It appears to me that PIN8 selects for IAA based on shape complementarity within the binding pocket, yet this is not really spelt out in the main text.

We agree with this observation and have emphasized shape complementarity on line 73:

"Comparison of these substrates suggests that shape complementary plays a large role in recognition: E.g. the larger size of Indole-3-butyric acid (IBA) and the reduced ring system of 2-phenylacetic acid (PAA) both result in reduced transport currents. "

also line 123 as follows: " Taken together, the interactions between the transporter domain and IAA again emphasize that PIN8 selects for IAA based on shape complementary as also suggested by the SSM electrophysiology. "

In ED Fig. 2D they report the results of transport for a panel of auxin compounds, but interestingly not PAA (2-phenylacetic acid)? Wouldn't this nicely show how important the indole ring is for affinity, just as the reduced current for IBA shows that longer ligands are not recognised well? I felt this part of the analysis was too cursory and could be expanded to draw conclusions between the chemistry of the ligands and their ability to be transported.

We have now included PAA in our analysis and that data is included in Extended Data Figure 2D. Although PAA elicits a current response, it is much lower than IAA. This result is consistent with the idea of shape complementarity of the indole ring.

We have added this to the main text in line 73:

"Comparison of these substrates suggests that shape complementary plays a large role in recognition: E.g. the larger size of Indole-3-butyric acid (IBA) and the reduced ring system of 2-phenylacetic acid (PAA) both result in reduced transport currents. "

Minor comment - please add the chemical structures to this ED Figure.

We have added the relevant chemical structures to Extended Data Figure 2F.

- Line 72. The authors say that NPA is a classical competitive inhibitor, yet don't mention the effect on V_{max} . Indeed, the effect of V_{max} on any aspect of the study (inhibitors/mutants) is strangely missing from the data.

Without this parameter, it is not possible to say with certainty this is classical competitive from the kinetic data alone.

In this paper we use a SSM electrogenic charge assay (SSM-SURFER) to measure activity, in addition to oocyte export measurements with radiotracer (radiolabeled IAA). SSM-SURFER has been used in 132 publications so far by our count and has been described thoroughly in by Bazzone, Barthmes and Fendler (2017, *Methods in enzymology*, "SSM-Based Electrophysiology for Transporter Research"). Most recently, this method was used by the group of David Drew to study the Na^+/H^+ exchanger Nha2, that we now reveal is structurally closely related to PINs (Matsuoka et al., *Nature Structural and Molecular Biology*, 2022).

SSM-SURFER is an extremely sensitive method that gives a current read-out in response to electrogenic events at the surface of proteoliposomes. Electrogenic events occur (i) when a charged molecule is crossing the membrane (ii) when a substrate, which does not necessarily have to be charged, binds to the protein and this binding leads to a conformational change by which charged elements move relative to the dielectric field of the membrane or (iii) currents are shielded/neutralized by the substrate and (iv) any combination of these possibilities. Disentangling the different components of the current response is not trivial. The readout used is normally peak current, and although this is the standard in the field, interpreting such a current response is sometimes difficult. We deal with this ambiguity by referring to the read-out as "current response" and avoid specifying the physical basis of the response unless we are certain. In the case of IAA, transport is supported by radiotracer studies here and elsewhere (Zourelidou, Marhava and many others). In the case of NPA, the term K_i and binding is warranted based on published data (Abas 2021) and this study (Figure 1C). We can exploit the binding signal from NPA to get an apparent K_d for IAA (see new ED Figure 2C where NPA binding current is inhibited by different IAA concentrations, giving us the apparent K_d of IAA). In all other instances we use " K_m " since it is the half-maximal concentration calculated by fitting a Michaelis-Menten curve, albeit this parameter could also more generally be described as EC_{50} (concentration effective in producing 50% of the maximal response). Thus, only in the case of competitive studies we use " K_d " or " K_i " since in this instance the parameter was specifically targeted by the experimental protocol.

A limitation of SSM-SURFER is that V_{max} (really the maximal current, I_{max}) from different samples cannot be reliably compared, due to uncertainty about the number of proteoliposomes and consequently number of transporters participating in the assay. Thus SSM-SURFER does not allow for the construction of a classical Lineweaver-Burk plot to document competitive inhibition. To tone down our kinetic interpretation of this data, we have removed the wording “competitive inhibitor” from the manuscript in the context of activity assays. The main argument for competitive inhibition here is and was structural. The inhibitor binds in the substrate binding-site, thus competing with the substrate for binding.

The text now reads (line 69):

"As in oocyte assays, transport can be inhibited by NPA, which inhibits with a K_i of 1.8 μM , suggesting an affinity one order of magnitude higher than IAA (Figure 1C)."

- In connection with the point above, the authors do not much discuss the effects of the mutants in Figure 2D. This is very odd as surely this is rather interesting. For example, the authors state that "mutation of Ser146 have little effect on activity". I disagree, the K_m drops by half and displays a higher peak current than WT. I think this is very interesting.. what is changing here, the K_m or the V_{max} ...? Similarly, what happens when you increase the concentration of IAA for the mutants that display higher K_m values? Does the peak current change or not? This could tell you whether the mutants are kinetically restricted, i.e slower. Does the current trace tell you anything about the kinetics? This part of the analysis is very cursory and I feel there is a lot that can be determined from this data which is currently lacking.

We agree the current data on the mutants is very interesting, but we do not want to over-interpret the available data. We have changed Figure 2D and now show all current responses at 500 μM of all our experiments done using WT PIN8 across different proteoliposome preparations. We also present the K_m of the average of all wildtype recordings (replaced panel in Figure 1B) and explain this in a more detailed version of the material and methods section.

Noted first in line 65:

“ Transport was measured using capacitive coupling by solid supported membrane (SSM) electrophysiology, showing that PIN8 has relatively low apparent affinity for IAA with a K_m of $356 \pm 136 \mu\text{M}$ ($n = 4$) (Figure 1B, Extended Data Fig 2C).”

It now becomes evident that the K_m of S146A falls in the K_m range observed for Wildtype and also the current responses that appeared stronger (e.g. in S146) than WT are within the range of current responses we observed in WT (cf. Figure 1B).

At this point we do not have a detailed kinetic model of the transport cycle that would lead to a comprehensive explanation for the effects of the mutants and limited space prevents discussion of the various caveats involved with any associated speculation. We are preparing this for a future publication.

We now state in line 112:

“ Mutating either Asn117 and Gln145 to alanine abolishes transport supporting their importance (Figure 2D). Tyr150 mutants display mixed results: Y150F retains activity, affinity and sensitivity to NPA, whereas removal of the bulky side chain in Y150A results in very low activity and affinity (Extended Data Figure 9A, B). In contrast, mutation of Ser146 had no effect on activity (Figure 2D, Extended Data Figure 9A). ”

- The structure of the NPA bound state is also interesting. I assume this result, combined with the fact that NPA inhibits from the extracellular side of the oocyte, suggests that NPA binds from the apoplast, whereupon PIN8 adopts the inward open state and the system stalls in this position because NPA cannot unbind into the cytoplasm.

No, this is not the assumption. Our data does not support that NPA is bound at the apoplast and then transported by PIN8 before inhibiting the protein. NPA has been shown to bind only to the cytosolic side of the protein in previous work (Abas et al, PNAS 2021). This is now confirmed by our structural data. See also Figure 2 and 3. In oocyte assays it was shown that NPA crosses the membrane in the protonated state before losing the proton at the cytoplasmic pH and then binding to PINs in the inward open state. Whether this is the mechanism in planta or whether some plant membrane proteins can transport NPA into the cytoplasm is not known. In oocyte experiments in Abas, 2021 as well as this publication NPA is injected into the oocyte together with the labeled substrate.

Note that we observe a binding current response of PIN8 to NPA. This binding current can be competed for by IAA. In response to the reviewer #2 request, we now exploit this property to obtain an apparent K_d for IAA. We have added the K_d data in a new panel (ED Figure 2D) and in the main text at line 67: “We measure the K_d of IAA binding to be 39.9 μM (Extended Data Figure 2D). ”

However, the authors do not really address what holds NPA in the binding site. I feel here, SPA (scintillation proximity assays) could have been employed to more fully understand this mechanism. However, in the absence of a binding assay to probe the importance of the interactions, the authors could investigate what effects certain functional mutations have on the K_i. The Y150F/A mutants for example look rather interesting.

We note that NPA binding is not irreversible. The interaction interface to the binding pocket is described in the text and in Figure 3B, C. A comprehensive analysis of mutants and their effect when exposed to NPA will be the focus of future work. However, for this manuscript, we have now tested NPA sensitivity of all mutants that have detectable levels of IAA induced current (Figure 2D): WT, I51Y, N117A, I120Y, Y150F, V328Y and Y150A. Overall it seems that NPA does not have statistically significant effect on I51Y, N117A and Y150A mutants. V328Y is notable for producing extremely reproducible data points for unknown reasons giving a significant result despite very low initial current. I120Y also shows lowered transport that is still sensitive to NPA. Both of these residues are at the proline cross-over motif at M4 and M9 respectively. Y150F retain almost WT activity and is also significantly inhibited by NPA.

This data is now included in ED Figure. 9B and is mentioned in the main text e.g. line 112:

“ Tyr150 mutants display mixed results: Y150F retains activity, affinity and sensitivity to NPA, whereas removal of the bulky side chain in Y150A results in very low activity and affinity (Extended Data Figure 9A, B).”

Line 121:

"This is supported by the bulky I120Y and V328Y mutants that both reduce apparent affinity by interfering with substrate binding, but still retain NPA sensitivity (Figure 2D, Extended Data Figure 9A, B)."

Also in reference to the NPA bound structure line 140:

" Several new interactions are observed also to the scaffold domain, many of which are mediated by the naphthyl ring of NPA and likely unique to the larger, more complex NPA molecule (Figure 3B, C and Extended Data Figure 9A, B, C, D)."

In addition, to better evaluate binding affinities of inhibitors as NPA we have considered Isothermal titration calorimetry (ITC) and Microscale Thermophoresis (MST) as explained for reviewer #3 (point 3) below to obtain a better understanding of substrate and inhibitor binding.

In brief, after initial trials we conclude that expression levels in our system do not produce sufficient amounts of protein for ITC. For MST we found that NPA fluoresces at both green and blue wavelengths and thus interferes with the fluorescent label required for our setup. See reviewer #3 for more detail on this approach.

- Line 95. The authors show in ED. Fig. 7C that lipids have no effect on PIN8 but do not explain what the significance of this observation, if any? I think at least some context here would be helpful.

We agree this was not fully discussed. Study of lipid dependence was motivated by the observation of a lipid molecule contributing to the dimer interface in the structure, which is noted in the text on lines 93-97. We have added a sentence to expand on the issue of lipid dependence in the figure legend.

Line 359:

" This supports that PIN8 is not dependent on specific lipids for activity."

- In connection with the above, the authors also reveal an unexplained density on the cytoplasmic side of the transporter that appeared in all structures. One assumes then that this molecule must come from the yeast expression system. This raises to questions in my mind, which may or may not be troubling..

1. Is this site regulatory in nature? If so, is the molecule something that is present in the yeast cytoplasm that could mimic an endogenous regulator of PIN8 in planta? I assume the authors tried to model in IAA?

This density appears at the periphery of the structure within the hydrophobic core of the bilayer, where a variety of other densities are visible at a similar threshold and possibly attributable to the peptidic belt or associated lipids surrounding the protein. We highlighted this particular density

because it is consistently seen at the same location in all our structures and we are of the opinion that novel observations should be presented even if an explanation is not readily available. But we have no way to definitively identify either the source or the identity of this density. We can model in IAA, and while it is not a perfect fit, we cannot reject this possibility.

2. How confident are the authors that this molecule is not interfering with their kinetic analyses? For example, the authors state that the K_m is much lower than for PIN8 in vivo, could this be the reason? The site is similar in nature to the binding site, mainly hydrophobic with a single H-bond donor/acceptor (Thr288) - unless I misread the figure? Have the authors mutated residues in this site to assess the effect on transport? T288A for example? It certainly leaves open the possibility that all parameters may not have been fully accounted for in the analysis of the data.

This is an interesting point. However, from a structural point of view it seems unlikely. The pocket is occupied in both inward and outward states and, when superposed, there are no structural changes in that region. Although we see the density in all of our structures (detergent and peptidisc stabilized), we do not know if it is occupied during the functional assays in vivo or in the SSM-SURFER since sample preparation is different (protein is in a lipid bilayer). Thus, it would be difficult to support any speculation about the identity of this density or associated allosteric effects based on existing data.

As suggested we have generated the T288A mutant to probe this density and do not observe any change in kinetics, although we have not produced a structure to determine if this mutation affects the presence of the density. We have added this data to Figure 2D and ED Figure 9A.

The hydrophobic pocket surrounding this density has not been observed in other cross-over elevator mechanism structures and might be a candidate for future investigations. However, anything we say at this time would be pure speculation and unlikely to add anything substantive to the current story. The legend to ED Figure 7 has been revised to provide better context for this observation.

Line 354:

" An unidentified density was found in maps in a cavity towards to the cytosolic side. Mutating T288A did not affect activity (Figure 2D and Extended Data Figure 9A)."

- The last section of the paper is too speculative for me. I very much enjoyed reading the paper, but the authors must admit there is not evidence in this paper to support any mechanistic role for proton binding/movement between D75 and K79.

This is correct. In fact, we believe that protons are not needed for transport. However, we are reluctant to over-interpret the data by concluding strongly in favor of one or the other option. It seems either proton-driven or proton-free transport is possible at this point. On the one hand, data from both the SSM-SURFER and oocytes show that PIN8 can function independently of a proton/pH gradient both in vitro and in vivo. On the other hand, D75N and K79Q mutants completely abolish transport. We have edited the text to clarify our position on this important aspect of transport.

Line 191:

"Our data show that the negative charge of the IAA is sufficient for transport (Extended Data Figures 2E and 10). "

Line 196:

" Our data thus support a uniport mechanism for PINs, though we cannot definitely rule out proton antiport in vivo."

Line 218:

"We describe the molecular mechanism of auxin transport by PINs that can function independently of monovalent ions or protons"

I appreciate the structural similarities to other SLC systems, but this is presented and discussed as if the 'support site' is mechanistically important and validated, which it isn't. It is perfectly conceivable, in my view, that this site has evolved to make PIN8 independent of pH or Na⁺ gradients, i.e. convert an antiporter into a symporter and nothing more.

We completely agree and apologize that this point was not clear in the original manuscript. We find it likely (but not proven) that PINs are indeed an example of antiporters converted into uniporters. This is similar in concept to the 'build-in counter ion' concept proposed for proton P-type ATPases (described in Pedersen et al. Nature 2007). We believe the dramatic effect of the isosteric mutations D75N, K79Q confirm the importance of the support site and support the existence of a conserved mechanism relative to other symporters and antiporters employing a crossover based elevator mechanism, despite evolution of PIN8 to function without a Na⁺ or H⁺ gradient. This point is mentioned e.g. in line 218:

"We describe the molecular mechanism of auxin transport by PINs that can function independently of monovalent ions or protons"

I think the authors should address this in their rebutal and amend the text and proposed mechanism in Figure 4. Or very clearly say this part of the charge-diffusion mechanism is pure speculation at this point and isn't based on any data presented herein.

We acknowledge that this original suggestion about a 'charge-diffusion mechanism' was speculative and thank the reviewer for pointing us in the right direction on this issue. Feedback at a recent conference also supported the speculative nature of the "charge-diffusion mechanism" and we have now amended our model to instead include the contribution from two positive dipoles (each about +0.5 charge) on either side of the cross-over that would help dissipate the negative charge of the ligand carboxylate. This feature is observable in figure 2A (positive potential in the binding pocket) and now mentioned in Figure 2B and Figure 4 and at line 104: "IAA is bound with the carboxylate group oriented towards the crossover, and stabilized by the positive dipole of the two M4b and M9b helices. "

and also line 198:

"The inward facing conformation allows an ionized auxin molecule to enter the binding site between transport and scaffold domains. The negatively charged carboxylate group is stabilized by the

positive dipole of M4b and M9b, while being held in place by Asn117 and interacting with the support site through Gln145.”

Minor comments

- The water molecules in the main figures should be colored something other than red, which is difficult to discriminate from the ligand carboxyl group.

We agree the position between the water and the IAA oxygen is very hard to distinguish in Figure 2C. We have removed the water from this panel as it was not contributing to the information content of this figure. After much testing and consideration, we have decided to retain the red color for the waters in Figure 2B, Figure 3C and Extended Data Figure 9B. It is customary in the structural literature to display waters in red, and we believe this color helps the reader interpret the figures quickly and correctly.

- Drawing the hydrogen bonds would be helpful in the main figure as in the ED figures.

In panel B of Figure 2 we find that the hydrogen bonds interfere with the presentation, given the focus on the side chains that constitute the IAA binding site, the crossover and the support site. The hydrogen network is indeed relevant later and is explicitly shown in Figure 3C.

Referee #2 (Remarks to the Author):

Ung et al. report the structures of Arabidopsis PIN8 protein, a member of the auxin efflux carrier family that has been less well-studied than the “canonical” PIN proteins with long hydrophilic loops. However, transport assays have shown that the efflux activity is conserved among the entire family. Therefore, PIN8 is probably a reasonable proxy for PIN protein structure and function. The auxin field has eagerly awaited the first PIN protein structures. The authors have leveraged the use of Cryo-EM on PIN8 protein expressed in yeast to generate structures of the apo-form, as well as IAA-bound and NPA-bound forms of the protein. Through analyzing these structures, their similarities to Bile Acid / Na transporters, through biochemical analysis and transport assays and mutant characterization, the authors present a mechanism for IAA transport by the PIN8 protein, as well as for its inhibition by NPA. This is a truly groundbreaking finding, that will open many new avenues of investigation and helps rationalize prior findings, such as the dimerization of PIN proteins and binding of NPA to PIN proteins.

We thank the referee for the strong support of our work.

PIN8 has been studied much less intensively than canonical PIN proteins, and is not exposed to the same environments as canonical PINs. While canonical PINs are exposed to apoplast on the outside and cytosol on the inside, PIN8 is ER-localized, and is exposed to the ER lumen on the “outside”. This has bearing on how well findings can be extrapolated (see point 2 below), but the strong conservation of key residues in the PIN family gives confidence that the mechanisms are also conserved.

PIN8 has been reported to be primarily located in the ER membrane and we use this to our potential advantage (see reply at point 2 below).

We agree that the high sequence similarity strongly supports a comparative analysis of PIN proteins on the basis of PIN8. There is further support from e.g. Bennett et al. (Bennett et al., *Molecular Biology and Evolution* 31, 2014) showing that PIN8 is evolutionarily clustered with PIN3 and the other canonical PINs (Figure 1 in that paper), and it is not a distant cousin (in contrast to the more unusual PIN5).

We can readily reproduce part of their analysis, showing that PIN8 is in the same clade as PIN3, PIN4, PIN7 while the two other non-canonical PINs (PIN5, PIN6) are not. (Analysis done with MAFFT for MSA, BMGE for MSA trimming, and FastME for tree generation, bootstrap result from 500 trials shown). Other methods yield similar results). We have now included this dendrogram as ED Figure 2A.

While of major importance, there are some points of attention:

1. It is unfortunate that a manuscript of this kind, reporting such an important breakthrough in auxin biology, lacks any form of *in vivo* validation. The authors admittedly developed a very good oocyte-based assay for auxin transport in which mutants are tested, but this is far from the natural context, where ER-localized PIN8 mediates specific functions.

Actually, we only tested the WT protein in oocyte assays, where we compared PIN8 with PIN1 and showed not only auxin transport but also inhibition by NPA. Comparison of WT with mutants and transport properties of various ligands were done using SSM-SURFER capacitive coupling, which employs purified, reconstituted protein in proteoliposomes. The major advantage of the SSM-SURFER assay is that we can derive kinetic parameters of the isolated proteins free from derived effects found in an *in vivo* setting. This makes the mutant analysis much stronger since it becomes independent of trafficking and other cellular regulation. We have detailed the SSM-SURFER methods above in the reply to reviewer #1 at the bottom of page 1.

Given that there are *pin8* mutants with well-described phenotypes, it would have been relatively straight-forward to complement the *pin8* mutant with versions in which key residues are mutated. I believe this is critical to reach the full impact of this study, and to translate *in vitro* findings to the biological context.

While *in vivo* analysis is certainly a future development based on our findings it is beyond the scope of this manuscript to link specific *pin8* mutants to plant phenotypes. At this point, the focus of our work is to provide biochemical and structural characterization of PIN8 and to provide a framework whereby mechanistic insight can be generalized to all PIN proteins. PIN8 was selected due to its amenability towards purification and structural determination as a good model protein for the general family. An *in vivo* phenotypic analysis for PIN8

or even other relevant PINs will be very time consuming and presentation of these findings would not be possible given length limitations of Nature articles. In fact, *pin8* single mutant phenotypes are extremely mild. A reduction of 10% in the proportion of “WT looking pollen” (Ding 2012, *Nature Comm.*) and a similarly very mild reduction in lateral root density (Lee 2020, *Frontiers*). The phenotypes have only been addressed by these two groups, and their results have not yet been reproduced by others. Additionally, the weak phenotypes make it extremely challenging to spot subtle differences between complemented mutants, mutants and WT. Different homologues will be

required for robust phenotypic studies and this clearly would be a separate project. But our results certainly provide a solid foundation for such a study.

2. As indicated above, PIN8 is an ER-localized protein, and this has consequences for the milieu that the protein faces are exposed to. If the protein expressed in oocytes for functional assays is in fact in the plasma membrane (which would be required to get a functional read-out in a transport assay – but is not shown), the sensitivity to pH, uncouplers, ions, etc, is perhaps not reflecting its normal activity. This needs to be explicitly addressed.

This is in principle correct. We agree the biochemical analysis is not done in the native environment, and this could lead to behavior that would not necessarily be observed *in vivo* for a variety of reasons, some relevant and some not. We note this is a very broad caveat that can be raised toward all types of biochemical characterization of purified proteins as well as non-native experimental setups within molecular biology.

PIN1 has been used in oocyte assays as a positive control, which support our straightforward conclusion that PIN8 transports IAA in a cellular context, can be inhibited by NPA, and does not require phosphorylation for activation. PIN1 has been shown to be in the plasma membrane in oocyte assays previously (e.g. Marhava et al., 2018 Nature) supporting our assumption that both PINs are indeed located to the plasma membrane, as is needed to get a functional readout. Additionally, the oocyte assay is carried out at pH 7.4 in the bathing medium which is close to the measured pH in the ER (7.2-7.3) (Shen et al 2013 Mol. Plant). We are therefore convinced that the experimental setup matches the natural environment as well as can be achieved. Importantly, results from the oocyte assay lend confidence that auxin transport seen in SSM-SURFER assays are not artefactual, Conditions for the SURFER assays were chosen to be the same as the oocyte assay, both for consistency and to mimic the situation in the ER as closely as possible, without the complications and ambiguities of *in vivo* analysis.

We now explicitly added this in the main text on line 170:

“The low apparent affinity for IAA measured in proteoliposome assays is 5-500 fold lower than the physiological concentrations of auxin in plant tissues (0.1 - 10 μ M)²⁰. While we cannot rule out experimental artifacts, this implies that distinct functions of Arabidopsis PINs arise from differing localization, abundance and autoinhibition properties rather than direct modulation of substrate affinity³.”

3. The authors explored the specificity of transport mediated by PIN8 in heterologous assays, and find that 2,4-D and NAA are about as effective as IAA in triggering a current in oocytes. This needs explanation, given that 2,4-D is used as a non-transportable (i.e. non-substrate) auxin *in vivo*.

There are multiple reports on the transportability of 2,4-D. To our knowledge it is not clear at this point if 2,4-D is a substrate for PINs. Several studies describe transport of 2,4-D *in vivo* which can also be stimulated by IAA (e.g. Hertel and Flory 1968 Planta “Auxin movement in corn coleoptiles”). We are aware that other studies (e.g. Delbarre et al. 1996, Planta) state that 2,4-D is not secreted by a carrier. One limitation could be that in the latter study used tobacco cells, and the identity of PINs was not known at the time; the suspension cells may just not express PINs. Nevertheless, these two contradicting examples illustrate that it is not firmly established that 2,4-D is not transportable in

vivo. In fact, a recent study found that 2,4-D resistance in radish was tightly correlated with actual transport of 2,4-D. The authors applied NPA and showed that this led to reduced export of 2,4-D from treated leaves (Goggin et al., 2016, JExBot). The authors concluded that ABCB transporters would transport 2,4-D since they assumed that NPA was an ABCB-specific inhibitor. In the light of newer findings (Abas et al. 2021 PNAS), and the observations in this study, we believe these data actually demonstrate that 2,4-D may very well be a transportable auxin in vivo and that PINs are likely to be the relevant carrier protein. We observe that 2,4-D elicits a current response in SURFER assays with PIN8, but we have not yet investigated whether this response reflects only binding or a binding/transport current, but the current is similar in shape and range to IAA, suggesting transport (binding would be expected to produce a faster, more transient response). At this point we know that the substance and the transporter interact which is consistent with its nature as an auxin. Whether 2,4-D is actually also transported will be investigated in the future.

In brief, our data shows that isolated and purified PIN8 in a proteoliposome can elicit a current response when exposed to 2,4D. This is what we now state in line 70.

" We screened a number of additional substrates for PINs (Extended Data Figure 2F) and found that IAA analogues, e.g. Naphthaleneacetic acid (NAA) or the herbicide 2,4-Dichlorophenoxyacetic acid (2,4-D), elicited a current response in PIN8 whereas uncharged auxins as well as some endogenous auxins did not

4. The authors identify NPA-interacting residues. Some of these are mutated and tested for activity (Figure 2D), but whether these proteins are still sensitive to NPA is not assayed. This would be important to conclude on the mechanism of NPA action.

We agree and we have now tested NPA sensitivity of all mutants that have detectable levels of IAA induced current (Figure 2D): WT, I51Y, N117A, I120Y, Y150F, V328Y and Y150A. Overall it seems that NPA does not have statistically significant effect on I51Y, N117A and Y150A mutants. V328Y is notable for producing extremely reproducible data points for unknown reasons giving a significant result despite very low initial current. I120Y also shows lowered transport that is still sensitive to NPA. Both of these are at the proline cross-over motif at M4 and M9 respectively. Y150F retains almost WT activity and is also significantly inhibited by NPA.

This data is now included in ED Figure. 9B and mentioned in the main text e.g. line 113:

" Tyr150 mutants display mixed results: Y150F retains activity, affinity and sensitivity to NPA, whereas removal

of the bulky side chain in Y150A results in very low activity and affinity (Extended Data Figure 9A, B)."

Line 121:

" This is supported by the bulky I120Y and V328Y mutants that both reduce apparent affinity by interfering with

substrate binding, but still retain NPA sensitivity (Figure 2D, Extended Data Figure 9A, B)."

Also at the NPA bound structure line 138:

" Several new interactions are observed also to the scaffold domain, many of which are mediated by the naphthyl ring of NPA and likely unique to the larger, more complex NPA molecule (Figure 3B, C and Extended Data Figure 9A, B, C, D). "

5. PIN8 has low affinity to IAA, when contrasted to predicted cellular auxin concentrations. The interpretation is not entirely clear to this reviewer. Is it possible that the polar accumulation of PIN proteins serves the role of generating high local protein concentrations to overcome low affinity for IAA binding? If so, how does this connect to the ER localization of PIN8?

It was a somewhat surprising finding, but we have now done wt analysis a number of times with different liposome preparations with consistent results. To better reflect the variation observed we now list the K_m when first reported in the text as the mean of four different preparations each consisting of at least two different experiments which consisted of three technical reps with Fig 1B showing a representative trace.

Line 65:

"Transport was measured using capacitive coupling by solid supported membrane (SSM) electrophysiology, showing that PIN8 has relatively low apparent affinity for IAA with a K_m of $356 \pm 136 \mu\text{M}$ ($n = 4$) (Figure 1B, Extended Data Fig 2C). "

We also have conducted an experiment to assess the K_d of IAA which is an order of magnitude smaller but still higher than endogenous IAA concentrations (discussed elsewhere).

The suggested idea of locally high protein concentration is an intuitive possibility, but remains to be demonstrated. The fact that the cell biology and particularly the plasma membrane occupancy of PINs is very tightly regulated seems to point in the direction that transporter abundance is in fact a way to regulate transport rates by transporter number. However, experiments to follow up on this concept are beyond the scope of the current manuscript.

We address this observation in the discussion in the manuscript, line 170:

"The low apparent affinity for IAA measured in proteoliposome assays is 5-500 fold lower than the physiological concentrations of auxin in plant tissues ($0.1 - 10 \mu\text{M}$)²⁰. While we cannot rule out experimental artifacts, this implies that distinct functions of Arabidopsis PINs arise from differing localization, abundance and autoinhibition properties rather than direct modulation of substrate affinity³."

6. It is interesting that PIN8 shows reasonable structural similarity to Bile Acid / Na transporters. Given the low sequence homology, it is perhaps not strange that this was previously undetected. With this knowledge however, it would be interesting to see how well PIN protein structure could be modeled using these templates.

We indeed made homology models in the earlier days of our structural analysis. However, we find that alpha-fold, with its more sophisticated machine learning algorithm, does a better job than homology modeling at predicting PIN structure. Given the almost non-existent homology with the bile acid transporters, failure of homology modeling is not particularly surprising. Nevertheless, now

that the structures reveal an empirical structural homology, comparison of these and other similar families are very helpful to our interpretation of how PINs function. See also Extended Data figure 8.

7. Likewise, the option of hetero-dimerization (or oligomerization) of PIN proteins is mentioned. It would be interesting to use modeling/docking to explore further if this is realistic.

We agree that this is an intriguing idea and worth mentioning in the manuscript as presented. However modeling of this would be highly speculative at this point and would warrant extensive experimental verification well beyond the scope of this paper.

Minor issues:

1. The term “morphogen” has a very specific connotation, and there is no convincing or conclusive evidence that

auxin acts as a morphogen

We have changed this word at line 17:

"Auxins are central hormones in all plants and controls virtually all aspects of growth and development."

and also line 32:

"Auxins are a group of hormones that regulate virtually all growth and developmental processes in plants."

2. “Most auxin effects are due to polar auxin transport” sounds strange...it is the actual response that triggers

cells to at, so at best, PAT is involved in these, or controlling these.

We have edited the abstract and removed this sentence as it was superfluous.

3. It would be helpful to describe the screening process that led to the selection of PIN8 for detailed

investigation.

We have added this to the methods section line 389.

Referee #3 (Remarks to the Author):

Auxin is a group of phytohormone and has critical roles in virtually all aspects of plant development. IAA is the most common naturally occurring of auxin. PINs are a family of membrane proteins that transport auxin across cells and tissues. Many studies have been directed at the developmental roles of PINs, very limited knowledge is available on their structures. The current manuscript reports biochemical and structural characterization of the short PIN protein Arabidopsis PIN8 (AtPIN8). Electrophysiological data showed that AtPIN8 was constitutively active in transporting auxin. The authors then solved cryo-EM structures of AtPIN8 in apo-, substrate IAA- and the inhibitor NPA-bound AtPIN8 forms. The structures showed that AtPIN8 contains the scaffold and transport

domains and forms a dimer. The structures revealed that Apo- and IAA-bound AtPIN8 adopt an outward-open conformation, whereas the NPA-bound AtPIN8 assumes an inward-open conformation. Structural comparison indicated that NPA binding causes rotation of about 20 degrees of the two transport domains of the AtPIN8 dimer. Data from mutagenesis analysis provided further evidence for recognition of IAA and NPA by AtPIN8. DALI search identified several families of secondary active transporters as close structural homologs of AtPIN8. Interestingly, these proteins are believed to function via an elevator type transport mechanism in which the transport domain undergo striking conformational changes during transport. The manuscript was well written and the structural data are solid. The results presented in the manuscript are significant for understanding the transport mechanism of auxin by AtPINs.

We thank the referee for the support of our work.

However, a number of issues need addressing before it can be accepted by Nature.

1. Based on the cryo-EM the structures in the manuscript, can the data presented in Extended Data Fig. 2D be explained? For example, 2,4-D differs from and 4-CPA only in one chloride atom. However, AtPIN8 displayed a much higher toward to the former than toward the latter.

This touches on a very interesting point. As also pointed out by Reviewer #1, we believe most of our data in ED Figure 2F can be explained by shape complementarity and that this is a major factor in substrate selectivity. In the example cited by this reviewer, the additional chloride atom would be expected to make only minor changes to the shape of the ligand, but there will be additional effects on the electro-negativity of the aromatic ring that could have more subtle kinetic effects on k_{on} or k_{off} that we are not able to measure with our current methods. Based on current structural information, shape complementarity does not seem sufficient to explain all of the nuances of ED Figure 2F. Note that in ED Figure 2F the current responses reflect 'peak current' as measured by SSM-SURFER. Although this is standard in the field, interpreting such a current response is non-trivial, as detailed in our response to Reviewer #1 on page 1, and it is possible that the change in overall electrostatic profile of 2,4-D by the addition of an additional Cl atom will change the current generated per molecule transported. Thus the results in ED Figure 2F should be evaluated with some care since the effect of different chemical molecules are measured. We have added the molecular structures of the ligands to this figure to help the reader assess their differences.

More work will be required to more fully elucidate the basis for substrate selectivity, but for the time being, we have added a line to the main text to acknowledge the uncertainties.

Line 323:

"Current response of substrates indicated with * differed significantly (One-way ANOVA followed by Dunnett's multiple comparisons test) from IAA, indicating that they are likely not substrates for the transporter, but we note that different chemical molecules have different electrostatic potentials and this can also have an influence of the observed current. "

We have also added this to the main text in line 73:

"Comparison of these substrates suggests that shape complementary plays a large role in recognition: E.g. the larger size of Indole-3-butyric acid (IBA) and the reduced ring system of 2-phenylacetic acid (PAA) both result in reduced transport currents."

2. The authors concluded that the negative charge of IAA is sufficient for AtPIN8-mediated transport. This is consistent with the on the chemiosmotic model of auxin transport. However, changes in pH only slightly affected the activity of transporting IAA by AtPIN8 (Extended Data Fig. 2D). Please make comments on this.

Given that the pKa of IAA is 4.7, this ligand will be negatively charged under all of the conditions that we studied. pH dependence was studied to evaluate the possibility of proton coupled transport. The modest pH effect that we observed argues against proton coupling and, as the reviewer states, support the chemiosmotic model of auxin transport, which we have mentioned in the manuscript on line 160-161. The decreasing activity below pH 5.5 is modest, but is consistent with the idea that unprotonated, charged ligand is preferred as a transport substrate, but this effect is hard to evaluate with the SSM-SURFER which relies on electrogenic transport i.e., a charged substrate. Additionally, changing the pH may also affect protein allostery in a more general way, thus affecting turnover. Nevertheless, our observations with SSM-SURFER are consistent with the oocyte data shown in ED Figure 10C, which show a lower transport by PIN8 at pH 5.5 compared to 7.4.

3. The IAA-binding affinity of AtPIN8 is extremely low, about 500-fold lower than the physiological concentrations of auxin. The authors have to verify the IAA-binding activity of AtPIN1 using other biophysical methods like ITC.

The IAA-binding activity of PIN8 (we assume the referee means AtPIN8 and not AtPIN1 here) is indeed of very high interest for a more comprehensive understanding of PIN function.

To better evaluate binding affinities, we have considered Isothermal titration calorimetry (ITC). Unfortunately, after initial trials we conclude that expression levels in our system do not produce sufficient amounts of protein for this method.

ITC measure minuscule changes in temperature as a ligand is titrated into a sample. It is therefore of the utmost importance that the buffer of the substrate and the buffer of the protein as well as the buffer in the reference cell are exactly identical. At the same time the signal is proportional to the affinity between substrate and the protein making low affinity interactions very challenging to measure even in the best of systems.

It is well known that ITC is normally not used for membrane proteins because the detergent environment of the sample cannot be matched in the reference cell; the resulting discrepancy will generally mask any signal that might be obtained from binding. However in favorable cases a weak signal can still be obtained. This is only possible if the protein is expressed at extremely high levels such that concentrating the sample at any point is not necessary; otherwise, sample concentration will also concentrate the detergent micelles and lead to buffer mismatch. This in turn requires SEC fractions of at least 5 mg/ml and preferably above 10 mg/ml combined with high μM affinity. We have neither in the case of PIN8, and therefore ITC is not realistic.

As an alternative, we have extensively tested Microscale Thermophoresis (MST); We tested MST using both the inhibitor NPA and the substrate, IAA, but after repeated attempts we conclude that it

is not possible to extract a signal from PIN8. We only observe a change in the signal at very high IAA concentrations. An SD-test (<https://nanotempertech.com/nanopedia/sd-test/>) show that this is a non-specific effect likely caused by inner filtering. MST measures changes in thermally induced tumbling rate of molecules, and we speculate that the binding of substrate to the protein does not sufficiently change the hydration or overall shape of the molecule to produce measurable differences. This is supported by the cryo-EM data that shows that apo-PIN8 and IAA-PIN8 are very similar in shape.

In summary, conventional methods for measuring binding affinity, ITC and MST, are not suitable for this system, so we have not been able to compare the K_m from transport assays with direct binding. Nevertheless, we have addressed this question from a novel angle using the SSM-based proteoliposome assay. Given that NPA can inhibit IAA transport, and that NPA elicits an observable binding current, we can have the proteoliposome exposed to a constant IAA concentration in the non-activating and activating buffer and then add NPA. We can now directly observe the reduction of the NPA binding current as NPA and IAA compete for the binding and this gives an approximation of the ' K_i ' of IAA (i.e. how much IAA is needed to inhibit NPA binding). This number will be equivalent to an apparent K_d of IAA.

We have now added this data (ED Figure 2D), and can show that with this approach the K_d of IAA binding is $\sim 40 \mu\text{M}$.

We have added this to the manuscript at line 67:

"We measure the K_d of IAA binding to be $39.9 \mu\text{M}$ (Extended Data Figure 2D)."

The low affinities are surprising, but as discussed in the main text, the numbers appear solid and may point to a mechanism by which transport rate is controlled by protein abundance, as described in the response to reviewer 2 above for point 5.

4. The major issue the reviewer had with the manuscript is about the proposed model on auxin transport by PINs (Fig. 4). It is true that the elevator mechanism has been suggested for some transporters. However, other alternating mechanisms like rocker switch and rocking bundle were also proposed for transporters. Can these two mechanisms be excluded by the data presented in the manuscript? The elevator-like model does not provide an explanation of why IAA is a substrate but NPA is an inhibitor of PINs. Evidence for the elevator mechanism of PIN-mediated auxin transport is that AtPIN8 bears structural similarity to some secondary active transporters. Structural comparison did reveal striking conformation differences between the apo-/IAA- and the NPA-bound forms of AtPIN8. While not described in the manuscript, I assumed that the authors used the transport domain as template for the structure alignment. However, striking conformation differences between the apo-/IAA- and the NPA-bound forms of AtPIN8 similarly exist in the scaffold domain if the transport domain is used as template for structural alignment. To the reviewer, this is more possible and even likely for at least for reasons. First, the major IAA-/NPA-binding site is located at the X-shaped crossover; second, such alignment puts IAA and NPA in a similar site around the crossover. IAA binding to the inward-open (NPA-bound) form of AtPIN8 generates steric clashes with Ile50, triggering conformational changes in the scaffold domain and leading to the outward-open

state. In contrast, NPA binding is unable to trigger conformational changes in scaffold domain, blocking AtPIN8 conversion into an outward-open state.

Our line of argumentation was perhaps not clear in the manuscript and we have revised the the Discussion to better explain our conclusions.

Our central argument goes along these lines: It is of course impossible to completely exclude other structural models for transport, but we believe that the preponderance of evidence favors an elevator mechanism for PIN8. To start, PIN8 has structural homology, i.e., shares its fold, with three other protein families (ASBT, SBTA, NAPA) that are well documented to operate by an elevator mechanism. In all of those cases, the scaffold domain (that generate the dimer interface both in PIN8 and in some of these other families) is considered to be the immobile element with the transport domain and substrate binding site translating relative to the membrane plane. Already in the absence of further information, this structural homology would support an elevator mechanism as the most likely model for PIN8. However, this model is also supported by comparison of PIN8 structures in the inward open and outward open states, which do indeed show a 5 Å displacement of the transport domain and substrate binding site relative to the scaffold domain, with substrate interactions being similar in the transport domain while the scaffold domain does not contribute. This is exactly the result expected for the elevator mechanism.

With regard to the alignments, we used the scaffold domain as the basis for alignment (not the transporter domain as the reviewer assumes), as has been done with these other related transporters. Given the dimeric complex formed by PIN8, this is the only thing that makes sense. If we were to align the transport domain of one monomer, then the conformational change would displace the scaffold domain and carry the other monomer out of the bilayer. This is clearly not energetically feasible. Even in the case of a monomeric transporter (e.g., ASBT), the scaffold domain is considered to anchor the protein within the membrane plane and provide an immobile reference point for the conformational change. These ideas are detailed in a recent review by Drew and Boudker (*Annu. Rev. Biochem.* 2016. 85:543–72). Drew and Boudker point out that scaffold domains often mediate oligomerization, though these concepts also apply to monomeric transporters, such as ASBT. This review goes on to explain that a defining feature of the elevator mechanism is that substrate binding and recognition is mediated by the transport domain and that the binding site then 'slides' against a fixed barrier created by the immobile scaffold domain (Figure 2 in the review, pasted here).

IMAGE REDACTED

This model also explains that NPA can inhibit transport because it makes specific interactions to the scaffold domain, NPA hinders the 'sliding' motion essential for transport. This is exactly what we observe in the structures and again lend support for the elevator model.

We agree that these points were not clear in the previous manuscript, and we have adjusted the text to address this. We have now removed the text related to a putative 'ring' motif as we find that this concept was likely an over-interpretation that caused confusion, and we now describe the inhibitory effect of NPA more clearly on line 142:

" Inhibition by NPA can thus be explained by two components: 1) stronger binding due to engagement of

additional residues from the scaffold domain and 2) the larger size of NPA that prevents transition to the outward state."

5. AtPIN8 forms a homodimer. But whether it is functionally required remains unclear.

We agree, and we now state so in the manuscript on line 208.

"Nevertheless, it is conceivable that the monomers operate independently and also that PINs could form hetero-

oligomers³³.

Reviewer Reports on the First Revision:

Referee #1 (Remarks to the Author):

The authors have addressed all my concerns/comments. I congratulate them on an excellent and informative study!

Referee #2 (Remarks to the Author):

While I remain of the opinion that the impact of a structural biology paper - no matter how convincing the in vitro biochemistry is - is far greater when the mechanisms reported (and amino acids involved) are in fact shown to operate in vivo, I can follow the authors' arguments for focusing this study only on the structure and in vitro biochemistry.

I am satisfied with explanations given and edits made.

Referee #3 (Remarks to the Author):

The authors have largely addressed my concerns. They now acknowledge that it is unknown whether AtPIN8 functions as a dimer. However, the the model was proposed based on dimeric AtPIN8. This should be explicitly pointed out in the manuscript.

Author Rebuttals to First Revision:

Reply to Referees.

(our answer in blue)

Referee #1 (Remarks to the Author):

The authors have addressed all my concerns/comments. I congratulate them on an excellent and informative study!

We thank the reviewer for the helpful suggestions that has improved the manuscript.

Referee #2 (Remarks to the Author):

While I remain of the opinion that the impact of a structural biology paper - no matter how convincing the in vitro biochemistry is - is far greater when the mechanisms reported (and amino acids involved) are in fact shown to operate in vivo, I can follow the authors' arguments for focusing this study only on the structure and in vitro biochemistry.

I am satisfied with explanations given and edits made.

We thank the reviewer for the helpful suggestions that has improved the manuscript. We agree future work include in planta experiments and this is currently underway.

Referee #3 (Remarks to the Author):

The authors have largely addressed my concerns. They now acknowledge that it is unknown whether AtPIN8 functions as a dimer. However, the the model was proposed based on dimeric AtPIN8. This should be explicitly pointed out in the manuscript.

We thank the reviewer for the helpful suggestions that has improved the manuscript. We agree the model of transport does not necessitate a dimer. This is stated in the manuscript (e.g. line 208) and also highlighted in figure 4 where the model shows only the monomer.